# Identification of quantitative trait loci (QTLs) for key cheese making phenotypes in the blue-cheese mold *Penicillium roqueforti*

Thibault Caron[1,2☖*], Ewen Crequer[1,3☖], Mélanie Le Piver[2], Stéphanie Le Prieur[1], Sammy Brunel[2], Alodie Snirc[1], Gwennina Cueff[3], Daniel Roueyre[2], Michel Place[2], Christophe Chassard[4], Adeline Simon[5], Ricardo C. Rodríguez de la Vega[1], Monika Coton[3], Emmanuel Coton[3], Marie Foulongne-Oriol[6‡], Antoine Branca[1,7‡], Tatiana Giraud[1‡*]

**1** Ecologie Systématique Evolution, IDEEV, Gif-sur-Yvette, France, **2** Laboratoire Interprofessionnel de Production—SAS L.I.P., Aurillac, France, **3** Univ Brest, Laboratoire Universitaire de Biodiversité et Ecologie Microbienne, Plouzané, France, **4** Université Clermont Auvergne, INRAE, Vetagro Sup, Aurillac, France, **5** Université Paris-Saclay, INRAE, Palaiseau, France, **6** INRAE, MycSA, Mycologie et Sécurité des Aliments, Villenave d'Ornon, France, **7** Université Paris-Saclay, CNRS, IRD, UMR Évolution, Génomes, Comportement & Écologie, Gif-sur-Yvette, France

☖ These authors contributed equally to this work.
‡ These authors jointly supervised the study.
* thibaultcaron864@gmail.com (TC); tatiana.giraud@universite-paris-saclay.fr (TG)

## Abstract

Elucidating the genomic architecture of quantitative traits is essential for our understanding of adaptation and for breeding in domesticated organisms. *Penicillium roqueforti* is the mold used worldwide for the blue cheese maturation, contributing to flavors through proteolytic and lipolytic activities. The two domesticated cheese populations display very little genetic diversity, but are differentiated and carry opposite mating types. We produced haploid F1 progenies from five crosses, using parents belonging to cheese and non-cheese populations. Analyses of high-quality genome assemblies of the parental strains revealed five large translocations, two having occurred via a circular intermediate, one with footprints of *Starship* giant mobile elements. Offspring genotyping with genotype-by-sequencing (GBS) revealed several genomic regions with segregation distortion, possibly linked to degeneration in cheese lineages. We found transgressions for several traits relevant for cheese making, with offspring having more extreme trait values than parental strains. We identified quantitative trait loci (QTLs) for colony color, lipolysis, proteolysis, extrolite production, including mycotoxins, but not for growth rates. Some genomic regions appeared rich in QTLs for both lipid and protein metabolism, and other regions for the production of multiple extrolites, indicating that QTLs have pleiotropic effects. Some QTLs corresponded to known biosynthetic gene clusters, e.g., for the production of melanin or extrolites. F1 hybrids constitute valuable strains for cheese producers, with new traits and new allelic combinations, and allowed identifying target genomic

**Data availability statement:** Reads and assemblies were deposited at GenBank under BioProject PRJNA1079247 for LCP06136, PRJNA1211835 for the eleven RxN offspring and PRJNA1216917 for the GBS data. Other available data are presented in S5 Table.

**Funding:** This study has been funded by the ANR-19-CE20-0002-02 Fungadapt ANR (to ECo), ERC Genomefun 309403 Stg (To TG), ERC Blue Proof of Concept (To TG), Fondation Louis D (French Academy of Sciences) grants (To TG), the LIP SAS and the ANRT (Association Nationale Recherche Technologie) (To DR and MP). The funders had no role in study design, data collection and analysis, decision to publish, or preparation of the manuscript.

**Competing interests:** I have read the journal's policy and the authors of this manuscript have the following competing interests: TC, MLP, SB, DR and MP were employed by SAS LIP (laboraroire Interprofessionel de Production), which produces starters for fermented food products, during the course of the study and therefore declare a competing financial interest. None of the other authors has any conflict of interest to declare, with the exception of TC employed by the LIP during his PhD, but not afterwards during his post-doc representing most of the present work.

regions for traits important in cheese making, paving the way for strain improvement. The findings further contribute to our understanding of the genetic mechanisms underlying rapid adaptation, revealing convergent adaptation targeting major gene regulators.

## Author summary

Understanding the genetic determinants underlying quantitative traits is crucial for our understanding of adaptation and for improving varieties in domesticated organisms. *Penicillium roqueforti* is the mold used worldwide for blue cheese making. The two domesticated cheese populations in *P. roqueforti* each display very little genetic diversity, but are highly differentiated. We therefore generated progenies between these populations despite reduced fertility. As expected, offspring displayed high variation in multiple traits important for cheese making, involved in aspect, flavor or food security, such as color, lipolysis, proteolysis and extrolite production. We detected genetic determinants for most of these traits, paving the way for generating new varieties. Several genomic regions impacted multiple traits, as often found in domesticated organisms. These findings further contribute to our understanding of the genetic mechanisms underlying rapid adaptation to new environments, likely involving master gene regulation impacting multiple genes.

## Introduction

Many traits of agricultural or evolutionary significance are quantitative and genetically complex, involving multiple genes that interact with one another and the environment [1]. Genomic loci showing diversity associated with quantitative trait variation are called QTLs, for "quantitative trait loci". The QTL approach has largely advanced our understanding of the genomic architecture of important traits, for example in crops and cattle [2–4]. Studies of the genomic architecture of adaptation in crops have revealed that traits involved in domestication were often controlled by only a few QTLs with pleiotropic effects, often being major regulators [5–16].

In the domesticated fungi used for food production, QTL mapping proved to be an efficient tool for identifying the genes affecting phenotypes impacting technological performances of industrial yeast strains (*Saccharomyces cerevisiae*), such as thermotolerance, chemical resistance, and phenotypes associated with the fermentation process, e.g., volatile compound and ethanol production [17–23]. In addition, traits have also been reported to be impacted in yeasts by aneuploidy [24] and by genomic rearrangements [25]. QTL mapping has also been successfully used in the button mushroom *Agaricus bisporus,* for identifying the genomic regions controlling color and spore number [26,27].

*Penicillium roqueforti* is the mold used for the maturation of all types of blue cheeses worldwide, responsible for the typical blue-veined aspect, and contributing to their specific flavor and aroma, in particular through high levels of proteo-lytic and lipolytic activities [28–30]. When this study was performed, four populations had been identified in *P. roqueforti*, two populations being used to produce cheeses (named Roquefort and non-Roquefort) and two populations being found in molded silage, lumber or spoiled food [31]. The two populations used for cheese maturation each show footprints of bottlenecks and a domestication syndrome, with phenotypes contrasting with those of the non-cheese populations and beneficial for several important aspects of cheese safety, appearance and flavor [31–33].

The Roquefort population, found primarily in Roquefort protected-designation-of-origin cheeses, exhibits traits bene-ficial for pre-industrial cheese production, *e.g.,* relatively slow growth in cheese medium and high spore production on bread medium, and produce high quantity and diversity of positive aromatic compounds in cheeses [31,32,34]. The other cheese population, named non-Roquefort, corresponds to a single clonal lineage, and is found in most types of blue cheese worldwide but rarely in Roquefort PDO cheeses. The non-Roquefort population displays phenotypes more suited for industrial cheese production, such as a faster growth in cheese, higher tolerance to salt and to lactic acid compared to other populations [31,33,35]. The non-Roquefort clonal lineage acquired large genomic regions by horizontal transfers, in particular two very large regions called *Wallaby* and *CheesyTer* [31,36], mediated by giant *Starship* mobile elements [37]. *Starships* are involved in adaptation in fungi domesticated for food fermentation [36,38], and also in plant pathogenic fungi [39,40]. The nucleotide diversity has been found very low in the non-Roquefort cheese population ($\pi = 0.000115$), about three-fold higher in the Roquefort population ($\pi = 0.000403$), and an order of magnitude higher in the lumber/spoiled food and silage populations ($\pi$ values of 0.00106 and 0.00160, respectively; [31,33]).

*Penicillium roqueforti* can secrete metabolites, also known as extrolites, including some considered as toxins (*e.g.,* Roquefortine C and PR toxin). However, these compounds are either absent in blue cheeses, or present at such low con-centration that they do not pose acute health hazards [41–43]. The non-Roquefort lineage has lost the ability to secrete mycophenolic acid (MPA), an immunosuppressant used for preventing transplant organ rejection [44], due to a deletion in a key gene (*mpaC*) of its biosynthesis pathway [33,45].

Identifying the genomic regions controlling the phenotypes contrasting between the four *P. roqueforti* populations thriv-ing in different niches has important fundamental implications for our understanding of the genomic architecture of adap-tation and also applied consequences for strain improvement by marker-assisted selection in progenies. Due to the clonal structure and low diversity of the domesticated cheese lineages [31], identifying the genetic determinants for cheese-making traits through phenotype/genotype association requires producing a recombinant population, which would also generate new allelic combinations and phenotypic diversity. Fortunately, sexual reproduction can be induced in *P. roque-forti* [46]. The two cheese clonal lineages carry opposite mating types and some cheese strains can be crossed, despite the general degeneration for sexual reproduction in the cheese populations [47]. Furthermore, the haploid life cycle of *P. roqueforti* enables carrying out association genetics in first-generation offspring.

Here, we aimed at generating new allelic combinations in cheese *P. roqueforti* populations and at identifying the genomic regions controlling the phenotypic differences between *P. roqueforti* populations. For this goal, we produced five sexual F1 haploid progenies by crossing six fertile haploid *P. roqueforti* strains from the four originally identified popula-tions. We studied in the progenies several phenotypes important for cheese making, *i.e.,* lipolysis, proteolysis, growth, color and the production of the main known extrolites: roquefortine C, PR toxin, with its intermediates eremofortins A and B, as well as andrastin A, (iso)-fumigaclavine A and MPA [48,49]. We analyzed 1,073 offspring based on single nucleotide polymorphisms (SNPs) obtained using genotype-by-sequencing (GBS). Because genotyping revealed the presence of the two parental alleles in some genomic regions in the otherwise haploid offspring, we also generated high-quality genome assemblies for the parental strains and eleven offspring, to compare the parental genomes and identify genomic translo-cations. We built genetic maps and ran QTL analyses to identify the genetic architecture of the phenotypes important for cheese making.

## Results

### Five fertile crosses involving at least one cheese strain

We chose the five most fertile crosses, based on qualitative assessment of the number of cleistothecia, among 17 trials involving in each case at least one cheese strain, chosen based on fertility data from a previous study [47]. Pairwise nucleotide identity estimates between parental strains ranged from 99.48% to 99.63% across crosses (Table 1). We identified the isolated spores that were actually recombinant ascospores and not asexual conidia by using 11 microsatellite markers in the Roquefort x non-Roquefort cross (386 spores were excluded out of 1620 tested spores). We focused on the cross between the two cheese populations, *i.e.,* Roquefort (R; LCP06136 strain) and non-Roquefort (N; LCP06133 strain), as it may be the most interesting for strain improvement in the cheese making context, and we isolated 387 recombinant offspring (Table 1). The other crosses each involved a cheese strain (either R or N) and a strain from a non-cheese population, either silage (S) or lumber/food spoiler populations (L). We isolated between 157 and 185 recombinant offspring for each of these four crosses (Table 1).

### Translocations between parental genomes

The comparison of high-quality genome assemblies revealed five large genomic chromosomal rearrangements between the parents in four crosses (LxN, RxN, RxS, RxL; Fig 1), which had impacts on genetic maps. These translocations ranged from 60 to 530 kb in size. The two silage parents and the non-Roquefort parent showed syntenic genomes between each other, suggesting that they kept the ancestral arrangement order. We could therefore infer that four of the

**Table 1. The five crosses performed using six strains of *Penicillium roqueforti* from different populations (R for Roquefort, N for non-Roquefort, L for lumber/food spoiler, S for silage/food spoiler) with opposite mating types (MAT1-1 and MAT1-2), with the number of isolated and phenotyped offspring, the number of marker obtained from GBS (genotyping-by-sequencing) and the average nucleotide identity (ANI) between parental strains. LCP: « Laboratoire de Cryptogamie, Paris ».**

| Cross | MAT1–1 | | MAT1–2 | | Number of ascospores isolated | Number of offspring GBS-genotyped | Number of offspring growth-phenotyped | Number of offspring color-phenotyped | Number of offspring lipolysis phenotyped | Number of offspring proteolysis phenotyped | Number of offspring phenotyped (mycotoxins) | Number of offspring used for map construction | Number of marker | ANI (%) |
|---|---|---|---|---|---|---|---|---|---|---|---|---|---|---|
| | Population | Strain | Population | Strain | | | | | | | | | | |
| RxN | Roquefort | LCP06136 | non-Roquefort | LCP06133 | 389 | 384□ | 307 | 311 | 387 | 382 | 274 | 358 | 2462 | 99.578 |
| RxS | Roquefort | LCP06136 | silage/food spoiler | LCP06043 | 157 | 157 | 157 | 157 | 157 | 156 | 0 | 148 | 2902 | 99.593 |
| RxL | Roquefort | LCP06136 | lumber/food spoiler | LCP06037 | 185 | 185 | 129 | 131 | 150 | 141 | 120 | 159 | 2943 | 99.630 |
| SxN | silage/food spoiler | LCP06059 | non-Roquefort | LCP06133 | 176 | 176 | 92 | 92 | 150 | 150 | 75 | 160 | 2860 | 99.484 |
| LxN | lumber/food spoiler | LCP06039* | non-Roquefort | LCP06133 | 171 | 171 | 91 | 91 | 150 | 151 | 0 | 165 | 2618 | 99.511 |

*The strain LCP06039 was found to be a mixture of two morphotypes ("dark" and "light"), which was detected after the cross had been performed.

□These offspring have been genotyped with indels, in addition to GBS, as all other offspring.

translocations (L1, L2, L3 and L4) occurred in the lumber population while the fifth one, R1, occurred in the Roquefort population (Fig 1).

The L1 and L4 regions displayed internal rearrangements between their copies in different genomic locations: the last 3' portion of each of the L1 and L4 regions in the parents with the ancestral arrangement was located in 5' of the L1 and L4 regions in the translocated parent (S1 Fig). This internal rearrangement suggests that the translocation occurred via a circular intermediate mechanism, with an insertion cutting site in each of the L1 and L4 regions different from their excision site (S1 Fig), as suggested to be the case for *Starship* mobile element movements [50]. As a matter of fact, we detected in the L4 translocation tyrosine recombinases (i.e., *Starship* "captains") and a DUF3723 domain, being all typical of *Starship* elements, using the dedicated *Starfish* pipeline [51].

Both parental alleles for the R1 region insertion were found together in 38, 31 and 24% of the RxN, RxL and RxS off-spring in GBS data, respectively. Only the R1 region was found in duplicate copies in these progenies, not the whole chromosome(s), indicating a lack of whole chromosome aneuploidy. The Oxford Nanopore long-read mapping depths confirmed the presence of two R1 region copies in the six analyzed offspring from the RxN cross with both R1 region parental

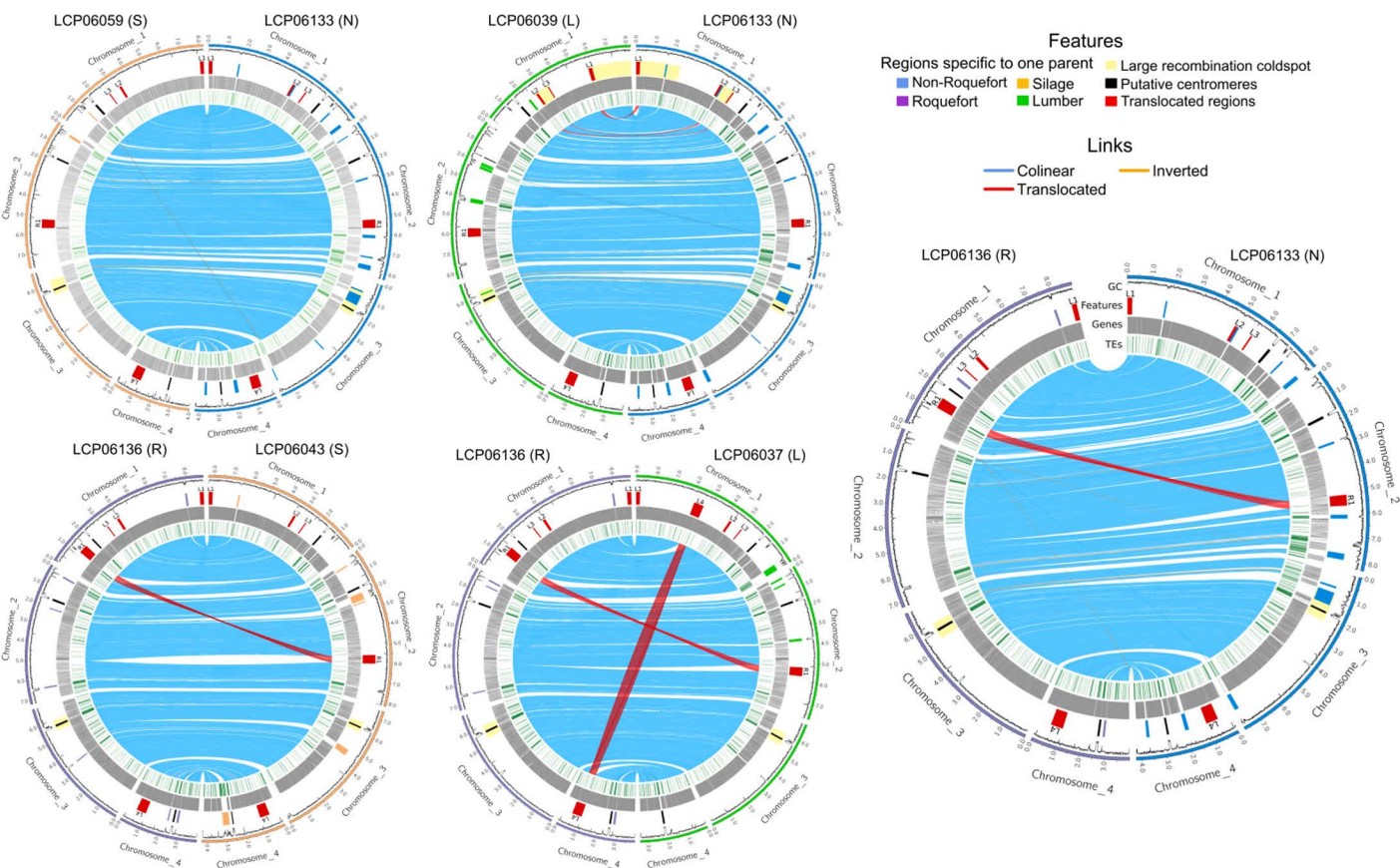

**Fig 1. Genome comparisons between parental strains for each cross.** Chromosomes are represented with colors corresponding to the population-of-origin of the strains: light orange, blue, purple and green for silage (S), non-Roquefort, (N) Roquefort (R) and lumber (L) populations, respectively. From outside to inside, the tracks represent the i) GC content, ii) translocated regions of more than 50kb between parental genomes in red (with their IDs, the first letter indicating the population-of-origin of the parent in which the translocation occurred), specific regions that are lacking in other genomes (*i.e.,* acquired horizontally transferred regions) with their color indicating the populations in which they are present, putative centromeres in black, and large recombination cold spots in light yellow, iii) predicted genes in gray and iv) transposable elements in green. Orthologs between genomes are linked one to each other, with blue, orange and red links, indicating synteny, inversions or > 50kb translocations, respectively.

alleles detected in GBS data and the lack of whole chromosome aneuploidy. Similarly, the long-read mapping confirmed that the five offspring with a single parental allele in GBS data in the R1 region carried a single copy of the translocated region in their genomes. The absence of this translocated region is likely lethal as we did not detect any offspring with no R1 copy, while some recombination events should have generated offspring with none of the two copies. We detected no offspring with both parental alleles nor an absence of the region for the translocations L1 to L4, indicating that both the duplication or absence of these regions lead to inviability.

We checked that the presence of the two R1 copies in haploid genomes was stable after culture and replication of 24 offspring of the RxN cross carrying the two parental alleles. After cultivating these 24 offspring from the RxN cross for one week on plates and transferring conidia from the edge of the colony to new plates 19 times, PCRs showed that the cultivated lineages still carried the two R1 parental alleles. We did not detect any RIP footprints (C-to-T repeat-induced point mutations) in the R1 regions of the six offspring with a duplicated R1 region and with high-quality genome assemblies. However, RIP footprints would only be expected in the following generation (F2), after a sex event involving genomes with the duplicated region, as RIP is known to occur during the short dikaryotic phase of sexual reproduction in ascomycetes [52].

**Genetic maps, recombination pattern and segregation biases**

We identified between 2,462 and 2,943 reliable markers in the progenies across the five crosses, evenly distributed along the non-Roquefort reference genome (LCP06133 strain). For constructing genetic maps, we excluded redundant markers, *i.e.,* those physically close and with identical segregation patterns in a given cross. We thus constructed the genetic maps with 1,448 markers for the RxN cross, and 676–785 markers for the other four crosses with fewer analyzed offspring (Table 2). The genetic maps for the five crosses displayed lengths between 775 and 913 cM, with a mean of 857 cM (Table 2). We detected four linkage groups in each genetic map, corresponding to the four chromosomes in the parental genome assemblies (Fig 2). The mean number of crossing-overs per chromosome was 2.15, with a maximum of 2.58 for chromosome 1, the longest chromosome, and a minimum of 1.25 for chromosome 4, the smallest chromosome. The mean recombination rate was estimated between 27.0 cM/Mb and 31.8 cM/Mb, with a mean of 29.8 ± 1.8 cM/Mb (Table 2). The LxN cross had a particularly low mean recombination rate, mainly due to the presence of two cold spots of recombination on chromosome 1 (indicated by a plateau in the plot of genetic map positions against genomic positions; empty rectangles in Fig 2), with local recombination rates of 6.7 cM/Mb and 3.1 cM/Mb, respectively (Table 2). The edges of the two cold spots corresponded to the translocated regions in the chromosome 1 of the LCP06039 strain from the lumber/food spoiler population (regions L1, L2, L3, Figs 1 and 2). This is likely due to the inviability of offspring carrying either zero or two copies of the translocated region because of recombination events between their insertion sites (Fig 1). Other regions with low or zero recombination rates corresponded to horizontally transferred regions only present in the reference genome, putative centromeres, other translocated regions and TE-rich regions (Figs 1 and 2).

In the five crosses, large regions presented segregation biases, with under-representation of alleles from the cheese populations (*i.e.,* either from the Roquefort or non-Roquefort parents; Fig 3). Such segregation biases could allow the purge of deleterious mutations in the cheese clonal lineages, but could also render the selection of valuable alleles that are in linkage with deleterious alleles in these regions more challenging. The segregation bias against the non-Roquefort parent allele in the second half of chromosome 4 (between 3.5 Mb and 4.2 Mb) was present in all crosses involving the non-Roquefort parent (Fig 3) and was very strong (>90% of the alternative allele in all three crosses). Such strong under-representation of alleles from the cheese population in all of these three crosses can be due to the lower fertility of domesticated populations [47], with likely deleterious alleles accumulated through clonal replication that lead to low viability of the ascospores carrying them. We observed another segregation bias with strong over-representation (> 90%) of silage parental alleles, in the RxS and SxN crosses, in chromosome 1 from 4.2 to 4.5 Mb (Fig 3). The under-represented alleles, that of the cheese parents, displayed no segregation bias in the other crosses involving the cheese parents (Fig 3),

suggesting the presence of either a selfish element (*e.g.,* a spore killer) or highly beneficial allele(s) for early growth in the corresponding region in the silage/spoiled food parents rather than a deleterious one in the cheese parent. This interpretation is supported by the over-representation of the silage alleles in two different crosses, and the lack of bias in the RxL and LxN crosses in this region, while similar biases would be expected in all crosses if the cheese parents had deleterious alleles in this region. The segregation bias with over-representation of Roquefort alleles at the end of chromosome 1 is likely due to the lack of the R1 translocated region that leads to offspring inviability. Symmetrically, the alleles from the other parent tended to be over-represented at the R1 locus on chromosome 2, but not significantly so. This suggests that the location of R1 may be more advantageous on chromosome 1 than on chromosome 2.

**Table 2. The five crosses performed using six strains of *Penicillium roqueforti* from different populations (R for Roquefort, N for non-Roquefort, L for lumber/food spoiler, S for silage/food spoiler), the linkage group and the total, the number of markers used for genetic map construction, the total length of the linkage group, the average and maximum spacing in cM between markers, the mean recombination rate in cM/Mb, and the reference chromosome and genome size in Mb.**

| Cross | Linkage group | Number of markers | Mean number of crossing over | Length (cM) | Average spacing (cM) | Maximum spacing (cM) | Mean recombination rate (cM/Mb) | Reference size (Mb) |
|---|---|---|---|---|---|---|---|---|
| SxN | L.1 | 213 | 2.74 | 274 | 1.3 | 8.8 | 32.21 | 8.51 |
| | L.2 | 242 | 2.49 | 249 | 1 | 12.8 | 29.97 | 8.31 |
| | L.3 | 189 | 1.97 | 197 | 1 | 8.8 | 25.51 | 7.72 |
| | L.4 | 94 | 1.19 | 120 | 1.3 | 8.8 | 28.47 | 4.22 |
| | overall | 738 | 8.39 | 841 | 1.1 | 12.8 | 29.25 | 28.75 |
| LxN | L.1 | 161 | 1.98 | 199 | 1.2 | 9.2 | 23.39 | 8.51 |
| | L.2 | 248 | 2.55 | 255 | 1 | 9.2 | 30.69 | 8.31 |
| | L.3 | 164 | 1.95 | 195 | 1.2 | 6.7 | 25.25 | 7.72 |
| | L.4 | 103 | 1.28 | 128 | 1.3 | 8.6 | 30.36 | 4.22 |
| | overall | 676 | 7.75 | 776 | 1.2 | 9.2 | 26.99 | 28.75 |
| RxN | L.1 | 466 | 2.78 | 279 | 0.6 | 7.3 | 32.79 | 8.51 |
| | L.2 | 435 | 2.48 | 248 | 0.6 | 8.7 | 29.85 | 8.31 |
| | L.3 | 378 | 2.22 | 223 | 0.6 | 9.6 | 28.88 | 7.72 |
| | L.4 | 179 | 1.25 | 126 | 0.7 | 8.7 | 29.89 | 4.22 |
| | overall | 1458 | 8.74 | 875 | 0.6 | 9.6 | 30.43 | 28.75 |
| RxS | L.1 | 191 | 2.74 | 275 | 1.4 | 11.7 | 32.32 | 8.51 |
| | L.2 | 242 | 2.47 | 248 | 1 | 13.9 | 29.85 | 8.31 |
| | L.3 | 227 | 2.55 | 256 | 1.1 | 12.4 | 33.15 | 7.72 |
| | L.4 | 115 | 1.33 | 133 | 1.2 | 7.5 | 31.55 | 4.22 |
| | overall | 775 | 9.10 | 913 | 1.2 | 13.9 | 31.75 | 28.75 |
| RxL | L.1 | 186 | 2.64 | 264 | 1.4 | 14.9 | 31.03 | 8.51 |
| | L.2 | 295 | 2.66 | 267 | 0.9 | 13.5 | 32.14 | 8.31 |
| | L.3 | 181 | 2.30 | 231 | 1.3 | 9.5 | 29.91 | 7.72 |
| | L.4 | 129 | 1.19 | 119 | 0.9 | 12.9 | 28.23 | 4.22 |
| | overall | 791 | 8.79 | 881 | 1.1 | 14.9 | 30.64 | 28.75 |
| Crosses mean | L.1 | 243.4 | 2.58 | 258.2 | 1.18 | 10.38 | 30.348 | 8.51 |
| | L.2 | 292.4 | 2.53 | 253.4 | 0.9 | 11.62 | 30.5 | 8.31 |
| | L.3 | 227.8 | 2.20 | 220.4 | 1.04 | 9.4 | 28.54 | 7.72 |
| | L.4 | 124 | 1.25 | 125.2 | 1.08 | 9.3 | 29.7 | 4.22 |
| | overall | 887.6 | 8.55 | 857.2 | 1.04 | 12.08 | 29.812 | 28.75 |

## Transgression and heterosis in phenotype distributions

We measured several phenotypes important for blue cheese production, *i.e.,* mycelium growth, colony color, lipolysis, proteolysis and extrolite production. We analyzed nine variables summarizing these traits in the five progenies, plus seven traits related to extrolite production in three crosses given time and cost limitations; the three crosses were RxL, RxN, SxN, choosing crosses involving at least one cheese population. Overall, these measures resulted in 66 trait distributions (9x5+7x3). Among these 66 distributions, 28 did not significantly deviate from normality, 10 best fitted a unimodal distribution and 30 a bimodal distribution (S2 and S3 Figs and S1 Table). None of the phenotype distributions were trimodal or amodal (S1 Table).

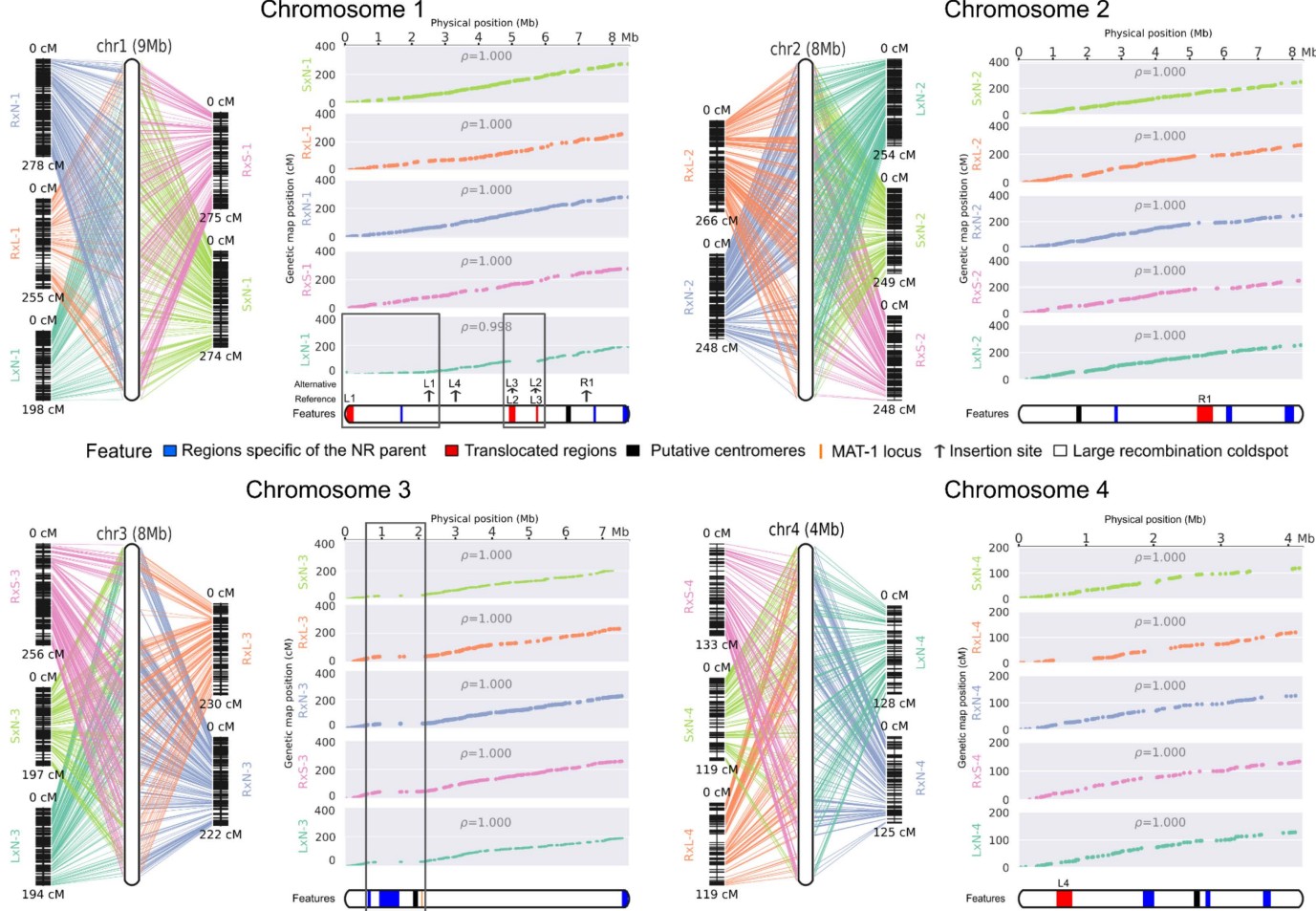

**Fig 2. Genetic and physical maps for the four chromosomes of *Penicillium roqueforti,* obtained by analyzing five inter-population crosses.**
Each panel corresponds to an individual chromosome, line graphs on the left represent linkage groups (one color per cross, with cross IDs as in Table 1; note that different parents were used for the silage and lumber populations between crosses); the lines link markers to chromosomes of the reference genome (LCP06133 non-Roquefort parent). In each panel, plots on the right represent the relationship between genetic and physical distance, with one plot per cross and the same color code as in left panels. At the bottom of each panel, the reference chromosome is represented, with filled black, red and blue rectangles representing the putative centromeres, the translocated regions and the horizontally transferred regions of more than 50kb when present, respectively. The red translocated regions are labeled as in Fig 2. The orange bar in chromosome 3 represents the mating-type locus position. The empty black rectangles indicate large recombination cold spots situated between translocated or parent-specific regions.

The distribution of the five progenies on the principal component analysis based on trait values (S4 Fig) shows that they represent overall a higher phenotypic diversity than each cross considered separately. A few traits appeared positively or negatively correlated (Fig 4D), some associations being expected, such as between color traits (*e.g.,* blue and red) or between the production of different targeted extrolites [34,53–55]. Other associations were more surprising, such as the negative correlation between proteolysis rate on the one hand, and green and hue level on the other hand (Fig 4D).

We found heterosis in the progenies, computed as the difference in mean trait values between the parents and their progenies (S2 Table). Out of the 66 phenotype distributions, 53 displayed significant positive or negative heterosis. We found negative heterosis in all crosses for the production of PR toxin (the most toxic *P. roqueforti* mycotoxin) and andrastin A, indicating that progenies produced on average less of these two extrolites than their parent mean. In contrast, we found positive heterosis in all crosses for MPA production, a less toxic extrolite. All tested phenotypes

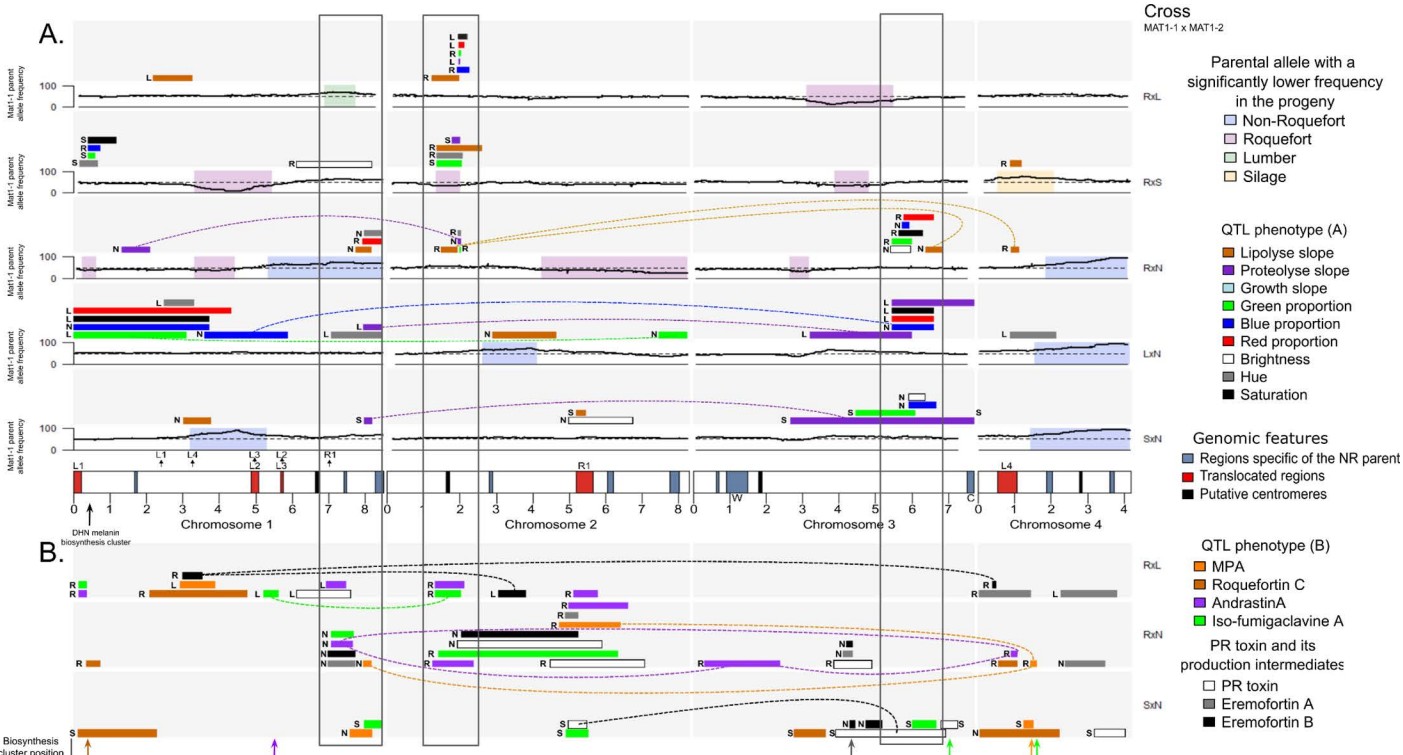

**Fig 3. Representation of identified quantitative trait loci (QTLs) and segregation distortion along the four chromosomes of *Penicillium roqueforti* in five progenies.** Only QTLs explaining more than 5% of the total variance are represented. The color of the bars indicates the considered trait while the letter indicates the parent having the alleles increasing the trait value. Dotted lines link the QTLs with significant interactions and color indicates the phenotype. The cross ID is indicated on the right, the first parent carrying the MAT1-1 mating type. **(A)** QTLs identified for phenotypes linked to lipolysis, proteolysis, growth and colony color for the five crosses (in lines, indicated on the right with the same code as in Table 1; note that different parents were used for the silage and lumber populations between crosses). Black curves represent the proportion in progenies of the allele from the parent carrying the MAT1-1 mating type. Transparent colored rectangles on curves indicate regions with significant segregation distortion, their color corresponding to the under-represented parental allele: blue, purple, green and yellow for non-Roquefort, Roquefort, lumber/spoiled food and silage/spoiled food parents, respectively. The x-axis represents genomic physical positions. At the bottom, the four chromosomes of the reference genome (LCP06133) are represented, with rectangles representing genomic features: large horizontally transferred regions specific to the reference genome in blue, and translocated regions in the reference genome in red (with same IDs as in Fig 2). The location of the horizontally transferred regions (W for *Wallaby* and C for *CheesyTer*) are indicated. The location of the dihydroxynaphthalene (DHN) melanin production cluster is indicated by a black arrow. The large vertical empty black rectangles indicate pleiotropic QTL regions, with effects on multiple traits. **(B)** QTLs identified for extrolite production (with colors corresponding to the different toxins), analyzed in three crosses (SxN, RxN and LxN). The locations of the gene clusters of the toxin biosynthesis pathways are indicated by arrows with the same color code (the PR toxin cluster being in gray, eremofortin A and B being intermediates).

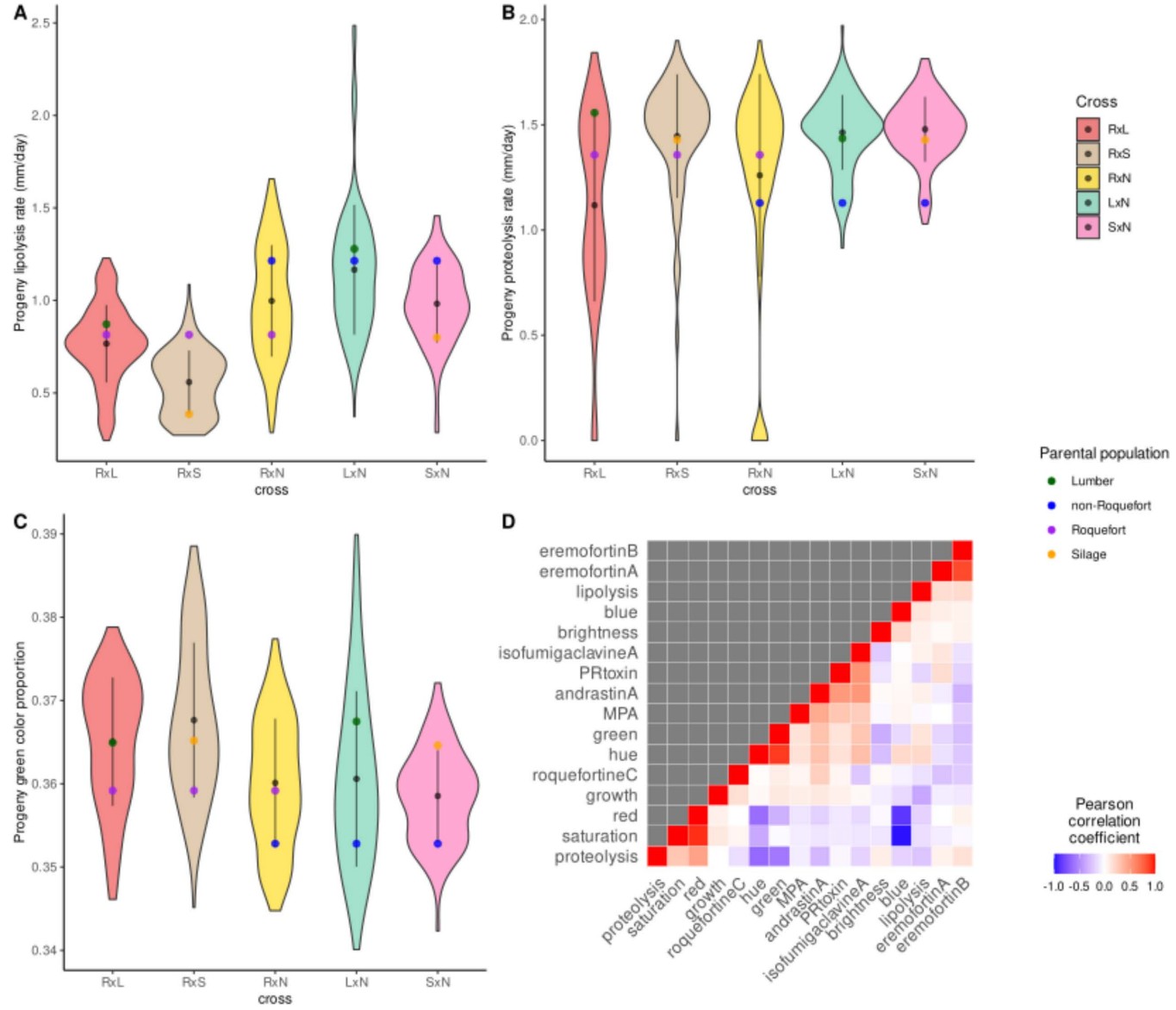

**Fig 4. Distributions of phenotypes in the five progenies in *Penicillium roqueforti*, for (A) the lipolysis rate (mm/day), (B) the proteolysis rate (mm/day) and (C) the relative green color channel of the progeny (RxL in red, RxS in beige, RxN in yellow, LxN in turquoise and SxN in pink).** The black point and line represent the mean and standard deviation, respectively, for each progeny. The colored points represent the parental values (lumber in green, non-Roquefort in blue, Roquefort in purple, silage in orange). **(D)** Pearson correlation coefficients, as a color gradient, between trait values across offspring and crosses.

exhibited both positive and negative transgressions across all crosses, *i.e.,* with some offspring displaying higher values and others lower values than either parent. We found a large proportion of offspring with low or no PR toxin production in the RxN and SxN crosses. We detected in all crosses negative transgression in roquefortine C production level (*i.e.,* the offspring mean was lower than the parental mean) and many offspring with no MPA production, which is promising for the selection of hypotoxinogenic strains.

## High values of phenotype heritability

Heritability can be estimated either as broad-sense heritability, accounting for total genetic variance (e.g., additive and epistatic), or narrow-sense heritability, only accounting for additive genetic variance. Narrow-sense heritability (h²) was estimated for all phenotypic traits across the five crosses (RxN, RxL, RxS, SxN and LxN) using a linear mixed model approach, based on the genetic relatedness between individuals and their phenotypes (S3 Table). Most traits analyzed exhibited relatively high narrow-sense heritability (h² > 50%) in at least one cross. Lipolysis and proteolysis rates, green proportion and extrolite production showed heritability estimates even above 70%, suggesting that these traits are primarily influenced by additive genetic effects, thus allowing selection in breeding programs. In contrast, the growth rate displayed consistently low heritability values, ranging from 9.6% (RxN) to 26.9% (SxN).

The broad sense heritability could be estimated for extrolite production. Estimates were high, ranging between 91.3% for roquefortine C and 99.6% for isofumigaclavine A, which is also promising for selection (S3 Table).

## Detection of QTLs with pleiotropic effects

We investigated QTLs for the nine traits associated with lipolysis, proteolysis, growth and color in all descendants, as well as for the seven traits associated with extrolite production specifically in the RxN, RxL and SxN crosses (S4 Table). We found QTLs for all tested phenotypes except for growth rate (Fig 3 and S4 Table), indicating that marker-assisted selection for these traits may be performed for strain improvement and diversification. We detected 123 QTLs across all phenotypes and crosses, 109 of which explained 5% or more of the phenotypic variance in a given cross (hereafter called "major QTLs"). We identified 58 major QTLs for the eight parameters related to lipolysis, proteolysis and color, across the five crosses, with 8–17 QTLs per cross, a mean of 1.3 and maximum of 3 major QTLs per trait and cross. For extrolite production, we identified 13–23 major QTLs across the three progenies analyzed for these traits, with a mean of 2.4 and a maximum of 5 major QTLs per extrolite and cross (S4 Table and Fig 3).

The identified QTLs and the direction of their effects were in agreement with the past occurrence of selection for color, proteolysis, lipolysis and extrolite production in cheese populations. We indeed identified QTLs impacting colony color, bluer colonies being associated with cheese parent alleles, and in particular in the genomic region containing the dihydroxynaphthalene-melanin biosynthesis cluster, known to be involved in melanin production (Fig 3). The cheese parent alleles were also both associated with slower proteolysis and faster lipolysis, in connection with firmer texture and longer cheese storage and the production of typical blue cheese flavors, respectively. We identified QTLs for the production of three extrolites, MPA, PR toxin and roquefortine C, the non-Roquefort alleles being associated with lower production levels (Fig 3). This is consistent with a selection for bluer color, more efficient lipolysis and less efficient proteolysis in domesticated lineages, as well as lower extrolite production levels.

Most of the major QTL regions presented pleiotropic effects. The 109 major QTLs were indeed not homogeneously distributed across the genome, with regions instead carrying clustered QTLs for different phenotypes (Figs 3 and S5). We considered as pleiotropic the regions presenting overlaps in QTL intervals of at least two different trait classes in a given cross (proteolysis, lipolysis, color and extrolite production). We identified five such regions, among which three regions displayed pleiotropic impact on the four trait classes, across the different progenies, and also within some progenies for multiple traits (empty vertical rectangles in Figs 3 and S5). For the pleiotropic region on chromosome 2, the Roquefort allele was associated with faster lipolysis and lower proteolysis. This QTL region displayed significant interactions for lipolysis with two other QTLs, in chromosomes 3 and 4; such interactions between loci in determining a trait value constitute positive epistasis. The same region also impacted color. For the pleiotropic regions detected in chromosomes 1 and 3, the non-Roquefort alleles were associated with slower proteolysis, faster lipolysis and bluer colony color (S4 Table and Fig 3). We further found epistasis for other types of phenotypic traits, particularly between different QTLs controlling proteolysis (S4 Table and Fig 3). QTL interactions had the same sign (positive or negative) across the

various crosses in some cases and different signs in other cases, indicating that epistatic interactions are dependent on the genetic background. Such epistasis could impair selection, as recombination breaks up allelic combinations, while specific allelic combinations are beneficial under positive epistasis.

We also identified QTLs for extrolite level production, including some outside of their known biosynthesis gene clusters, suggesting that these QTLs correspond to *trans*-acting regulators. For three extrolites (MPA, PR toxin and roquefortine C), we identified QTLs with intervals included in their biosynthesis cluster, with non-Roquefort alleles associated with lower production levels, consistent with selection for an hypotoxinogenicity in this population. The lower MPA production level associated with the non-Roquefort allele is likely due to the deletion in the *mpaC* gene that was identified in the non-Roquefort population [34,45]. We also found a QTL in a region including the PR toxin biosynthesis gene cluster, with lower production level of the PR toxin but accumulation of its production intermediates, eremofortins A and B, associated with the non-Roquefort allele at the PR toxin biosynthesis cluster. This suggests a disruption in the cluster, driving low production levels of the toxin and accumulation of its intermediates in offspring presenting the non-Roquefort allele, likely due to a premature stop codon in the ORF11 gene of the cluster [45]. Most of the QTLs associated with extrolite production were not located within their biosynthesis gene cluster, which suggests the implication of *trans*-acting regulators. These QTLs were often shared between different extrolites, suggesting a co-regulation of their pathways, which could facilitate the selection for low extrolite production in strains bred for cheese production. In particular, isofumigaclavine A production shared five of its eight major QTLs with andrastin A production, suggesting co-regulation of the production of the two extrolites. We detected QTLs for all tested extrolites, except roquefortine C, in the R1 region for its two insertion sites (Fig 3), with lower production associated with the presence of the R1 region. Such a QTL clustering suggests the presence of a master regulator of extrolite production in the R1 region. We, in fact, found an ortholog of a master regulator gene in the R1 region, namely the *srk1* gene, regulating osmotic and oxidative responses in fungi [56].

We detected an andrastin A QTL next to the *Wallaby* horizontally transferred region (HTR) in the RxN cross and a proteolysis QTL in the region containing the *CheesyTer* HTR in each of the SxN and LxN crosses (Fig 3). Multiple other phenotypes have previously been suggested to be controlled by these HTRs, such as milk sugar metabolism and interactions with other microorganisms [36], but the indel nature of these HTRs may render hard to detect their associations with phenotypes, as markers are lacking in one of the parents.

We also checked whether QTL regions encompassed other regulators known to be involved in extrolite production (*prlaeA*, [55], *pcz1*, [57], and *sfk1*, [53]), as well as other essential phenotypes in *P. roqueforti*, such as conidiation or growth rate. We located *pcz1* in the pleiotropic region of chromosome 2, encompassing QTLs for andrastin A production and color, melanin being involved in both color and conidiation [57]; *pcz1* could therefore be a candidate for the QTLs controlling andrastin A production and color via expression regulation. While no QTL interval overlapped with *prlaeA*, a QTL region controlling MPA production in one cross encompassed *sfk1*. However, this QTL region overlaps with a shorter MPA QTL present in another cross without *sfk1* in the confidence interval, which suggests no implication of *sfk1* in MPA production regulation.

The QTL intervals were too large to identify particular genes or functions beyond the horizontal gene transfers, biosynthesis gene clusters of the targeted extrolites or other *a priori* candidate genes. Indeed, with on average one gene every 3kb in the annotated Roquefort parental strain and a median QTL length of 760 kb, half of the QTL regions are expected to encompass more than 250 genes. We therefore only tried to identify candidate genes in the smallest QTL intervals explaining more than 20% of the phenotypic variance. We found an enrichment in genes involved in carbohydrate catabolism for a QTL associated with color in the pleiotropic region of chromosome 2 (g3621 to g3634), and a pepsin gene (endopeptidase, g5778) for a QTL associated with proteolysis in chromosome 3. Carbohydrate catabolism and pepsin are of paramount importance for secondary starters, on one hand, to use the remaining lactose residues to establish themselves early in the cheese matrix [28], and, on the other, to degrade caseins and therefore to participate in proteolysis [58].

## Discussion

### Variability in phenotypes in progenies, with high heritability, heterosis and positive transgression for most traits

We found a large phenotypic variation, transgression and heterosis in progenies for multiple relevant phenotypes for blue cheese production, *i.e.,* mycelium growth, lipolysis, proteolysis and extrolite production. The generation of new allelic combinations is particularly important in *P. roqueforti*, as the two main cheese populations, Roquefort and non-Roquefort, have lost most of their genetic diversities due to recent bottlenecks driven by strong selection in industrialization times [31]. These clonal lineages have degenerated in terms of sexual fertility and probably for other traits of interest for cheese making [47], and the use of progenies could purge accumulated deleterious mutations [59].

We found high values of heritability for most traits, in particular for toxin production, as previously reported for aflatoxin production in the fungus *Aspergillus flavus* [60], an opportunistic pathogen causing aspergillosis disease in humans. High values of heritability were also found for many traits in the yeast *S. cerevisiae*, in particular for traits important for beverage production [61,62]. High heritability was also reported for traits related to fitness in wild fungal species, such as the plant pathogens *Monilinia vaccinii-corymbosi* and *Rhizoctonia solani* [63,64].

We observed transgressive segregation in *P. roqueforti* progenies. Some offspring indeed displayed more extreme trait values than both parents in key traits such as color, metabolite production, as well as proteolytic and lipolytic activities. Transgression, where offspring exhibit extreme phenotypes, with values beyond those of their parents, has been documented in a wide range of organisms. For example, transgression has been leveraged in the yeast *S. cerevisiae* to enhance fermentation ability and improve morphological characteristics [65–67]. Transgressive effects have also been reported in crops like rice, sorghum, and lettuce for agronomic features [68–71] and in the button mushroom *A. bisporus* for yield-related traits and color [26].

### Translocations and segregation biases

The high-quality genome assemblies led to the identification of five translocations between the parental strains, having occurred independently in three lineages. These translocations and the resulting copy-number variation in some offspring could be one of the reasons why the fertility of crosses is low and variable among crosses in *P. roqueforti* [47]. In *S. cerevisiae,* non-reciprocal translocation can result in segmental duplication in offsprings, leading to genomic imbalances and reduced viability [72]. A similar reduction in offspring viability due to translocations has also been reported in *Neurospora crassa* [73]. We detected lower production levels for the targeted metabolites in offspring carrying two copies of the R1 region. These offspring may therefore be highly valuable for cheese making, especially as the presence and locations of the translocations were stable through multiple replications and culture events.

The L4 translocation appeared to have occurred via a circular form and encompassed domains typical of mobile *Starship* elements, indicating that additional *Starships* are present in *P. roqueforti* compared to the previously detected ones, shown to be involved in adaptation [36,37]. The L1 translocation, having also occurred by a circular intermediate, may also be a *Starship*, even if the *Starfish* pipeline did not detect the typical domains, but it can sometimes miss them. The other translocations may also be *Starships* or have occurred without being mobile elements.

Chromosomal translocations between domesticated lineages and their wild relatives, possibly affecting crosses and thereby breeding, have been reported in other fungi, for example in the yeast *Lachancea cidri* [74], but also in plants: soybean [75], wheat [76], oat [77], quinoa [78] and chili pepper [79]. These translocations sometimes provide beneficial phenotypes, such as fermentation and ethanol tolerance in yeast [74]. Structural variation has however been suggested to be mostly deleterious in domesticated lineages and to represent a cost of domestication, for example in rice [80] and grapevine [81].

## Crossing-over numbers per chromosome and recombination rate

In eukaryotes, the mean number of crossing-overs is often around two per chromosome, with variability across crosses [82], which is in agreement with our findings in *P. roqueforti*. Among fungi, the number of crossing-overs per chromosome is between 4 and 10 in ascomycetous yeasts, such as *Saccharomyces cerevisiae* [83] and *Komagataella phaffii* [84], above 15 in the ascomycete *Aspergillus fumigatus* [85], between 0.25 and 1.8 in the basidiomycete *Lentinula edodes* [86], and between 0.19 and 1.33 in another basidiomycete, *Hericium erinaceus* [87]. Thus, our estimates in *P. roqueforti* of 2.15 crossing-over per chromosome, on average, is among the lowest numbers of crossing-overs per chromosome described in ascomycete fungi. Our estimated recombination rate in *P. roqueforti* (29.81 cM/Mb) was also lower than the average reported in fungi of 48.68 cM/Mb [88], but closer to the average for eukaryotes (22.93 cM/Mb; [88]. Fungi indeed generally display high recombination rates compared to other eukaryotes, mainly because of their small genome size, given the negative correlation between genome size and recombination rate [88].

## QTL identification, strain breeding and maintenance of genetic diversity in domesticated populations

We identified QTLs for all phenotypes but growth. There were only a few major QTLs per trait and cross, which should facilitate strain breeding. The QTLs for color, lipolysis and proteolysis on the one hand, and for extrolite production on the other hand, clustered in the same genomic regions, suggesting pleiotropy, or clustering of genes with effects on these different pathways. The control of traits involved in domestication by only a few QTLs and their pleiotropy have been reported in animals and plants, often mainly due to major transcription regulators, human selection having unconsciously targeted "masterminds" [10], but also to the genetic determinant of important crop traits being clustered in genomes [9,15]. Here, we show that such pleiotropic regions are present in two independently domesticated lineages of *P. roqueforti,* with the same type of effect on the phenotype, suggesting evolutionary convergence [89] and selection of major regulators with pleiotropic effects. The pleiotropy of many QTLs can either facilitate or impair the selection for desired trait combinations. For example, antagonistic pleiotropy impaired industrial strain optimization in *S. cerevisiae* and *Schizosaccharomyces pombe*, because of a trade-off between stress resistance and ethanol production [90].

The discovery of additional, non-pleiotropic QTLs in *P. roqueforti* indicates the potential for a finer selection on individual traits without negatively impacting others. QTLs can serve as valuable markers for breeding programs, as previously done for improving fermentation traits in yeast [91]. Marker-assisted selection has been successfully applied to mitigate phenolic off-flavor or hydrogen sulfide production in *S. cerevisiae* [92] and for the rapid selection of the sporeless trait in the basidiomycete *Pleurotus pulmonarius* [93].

For strain breeding in *P. roqueforti*, further efforts could include integrating the recently discovered Termignon cheese population. The Termignon population is indeed genetically and phenotypically distinct from other cheese populations, likely representing remnants of an ancestral-like population of early domesticates [33]. This population thus offers a unique genetic reservoir for strain breeding and for enhancing diversity in cheese production. In general, more efforts are needed in agro-ecosystems for sustainable use of crops and domesticated micro-organisms, to maintain extant genetic diversity, and reintroduce diversity from wild populations or even closely related species [94–96].

## Materials and methods

### Progeny production

The production of progenies was obtained following the protocol in [46] with some modifications. Six *P. roqueforti* strains (Table 1) were selected for their fertility as observed in a previous study [46] and based on their phenotypic differences estimated from proteolysis and lipolysis measurements [31]. They were crossed in all possible pairs given their mating types, either MAT1-1 (three strains) or MAT1-2 (three strains) (Table 1).

For each cross, three Petri dishes were inoculated on an oat medium supplemented with biotin after sterilization (6.4 µg/L; [97]). On each Petri dish, 5 µL of an uncalibrated spore suspension was used to inoculate agar sections at opposite sites on the Petri dish, located perpendicular to the sections inoculated with aliquots of conidia of the opposite sexual type (Fig 5A; [98]). Petri dishes were sealed with food-grade plastic film and incubated upside down at 19°C in the dark. The contact zone between strains (Fig 5A) was examined with a binocular loupe and an optical microscope regularly for four to six weeks to check for the presence of cleistothecia (sexual structures). When cleistothecia were found, three to five of them were sampled, crushed with tweezers, and observed under an optical microscope with lactophenol cotton blue dye solution for ascospore detection (Fig 5B and 5C). We took the pictures with a Nikon DS-Fi2 camera, on an inverted optical microscope with apodized phase contrast (Nikon France, Champigny-sur-Marne, France). When ascospores were observed, all the cleistothecia of the corresponding Petri dish were collected in a 0.05% Tween-20 water suspension (P1379-100mL Sigma Aldrich, Saint Quentin Fallavier, France). Because cleistothecia are covered with conidia derived from asexual reproduction, we developed a treatment to kill them before releasing ascospores from cleistothecia. Approximately 500 µL of the cleistothecia suspension was pooled with 500 µL of a commercial 2.6% sodium hypochlorite solution and gently agitated for 20 seconds; then 1 mL of water was added to stop the sodium hypochlorite action. Cleistothecia were washed twice with one-minute centrifugation at 7,000 rpm, removal of the supernatant and resuspension in Tween 20. Then, we crushed cleistothecia with a sterile piston (Piston Pellet Eppendorf, Montesson, France) and filtered residues with a 40 µm filter (cell screen, Greiner, Les Ulis, France). Single-ascospore isolation was carried out with a dilution method on malt extract agar Petri dishes (MEA, [99].

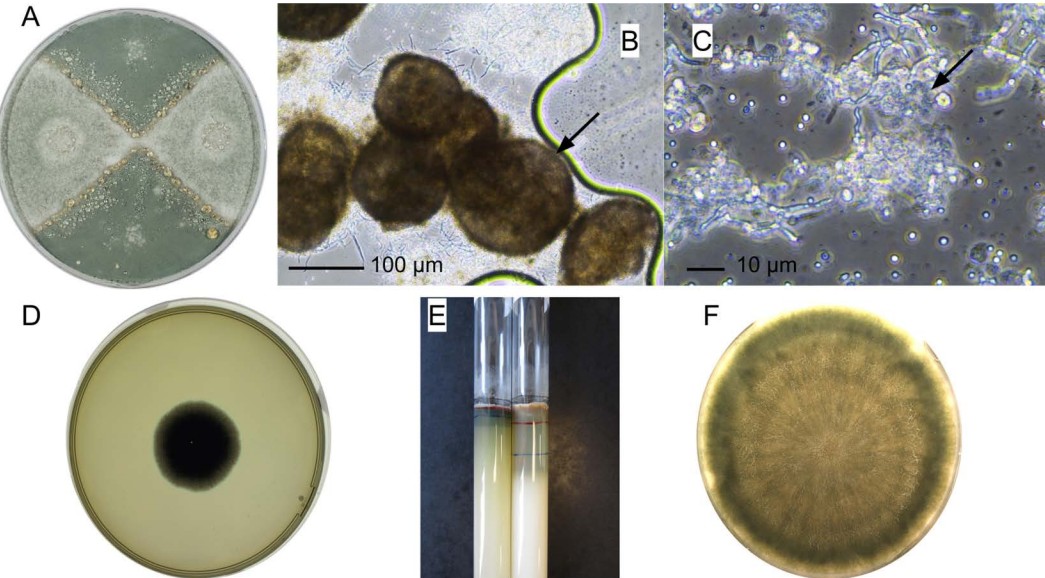

**Fig 5. Observations of *Penicillium roqueforti* crosses and examples of phenotypic trait determination. (A)** Petri dish with a cross between the LCP06136 (MAT1-1; Roquefort; at the top and bottom) and LCP06173 (MAT1-2; non-Roquefort; at left and right) strains. Sexual structures are formed at the contact zones between the two strains. **(B)** Cleistothecia (sexual structures of *Penicillium roqueforti*) shown by black arrow. **(C)** Asci containing ascospores shown by the black arrow. **(D)** Petri dish used for estimating colony growth rate, by counting the number of pixels occupied by the fungal colony (one offspring grown on malt medium for 120 hours). **(E)** Two lipolysis test tubes showing the lysis dynamics, for two different strains, with marks on the tubes indicating the limit of lipid medium degradation at different times (blue, red and blue marks, drawn at 0, 7 and 14 days, respectively). **(F)** Petri dish on which an offspring grew on raw sheep milk medium during 13 days for color measure.

## Recombinant offspring detection

We chose five fertile crosses for further analyses (Table 1). Genomic DNA was extracted from fresh mycelium and conidia suspension after single-ascospore isolation and growth for five days on MEA media using the Nucleospin soil kit (Macherey-Nagel, Düren, Germany). For checking that isolated spores were recombinant offspring and not asexual conidia, we used 11 microsatellite markers [46] labeled with fluorescence, out of which between five and nine were polymorphic depending on the cross. These markers were amplified with the Multiplex PCR kit (Qiagen, Les Ulis, France) using a touchdown program with an initial denaturation of 15 min at 95°C, 35 cycles of 30 s at 94°C, a decrease of 1°C every 90 s from 60 to 50°C, and 60 s at 72°C. The PCR program ended with a final 30 min extension step at 60°C. Genotyping by capillary fractionation electrophoresis was performed at INRAe Clermont-Ferrand (INRAe Platform GENTYANE, Clermont-Ferrand, FRANCE). The profiles obtained were analyzed with the GENEMAPPER v4.0 software (Applied Biosystem, Villebon-sur-Yvette, France) to detect recombination between parental genotypes. Only strains with recombinant genotypes were retained, *i.e.,* carrying alleles from one parent at some markers and alleles for the other parent at other markers.

## Genome sequencing, assembly and analysis

We generated long-read-based genome assemblies by sequencing the genomes of the LCP06136 parental strain, as well as of 11 F1 offspring, with Oxford Nanopore MinION technology with an R9 flow cell, a high-quality genome assembly of the other parental strain (LCP06133) being available [33]. We also generated genome assemblies for the other parents, based on Illumina sequencing and assemblies guided by the Oxford Nanopore assemblies of the LCP06136 and LCP06133 parents (S5 Table).

DNA was extracted from mycelium and conidia with the Nucleospin soil kit for the progeny and with the NucleoBond High Molecular Weight DNA kit (Macherey-Nagel, Düren, Germany) for the parental strain LCP06136, with mechanical disruption of about 30 mg of lyophilized mycelium with two tungsten beads (3 mm diameter) for 5 min at 30 Hz. The Nanopore library was prepared with the SQK-LSK109 ligation sequencing kit, starting with 1.5 μg DNA, and sequencing was performed in-house with a MinION combined with a MinIT (version 19.06.9), a 72 hour run, and with the Fastcalling basecalling algorithm. The genome of the LCP06136 strain was sequenced alone in a R9 flow cell, whereas the 11 F1 offspring genomes of the LCP06133 x LCP06136 cross were sequenced in a 12-plex with the Rapid Barcoding Sequencing kit SQK-RBK004. We assessed run quality using Porechop v0.2.3_sequan2.1.1 [100], and when necessary, demultiplexed the Nanopore raw reads with the default parameters.

*De novo* assemblies of the genomes were constructed from both Illumina and Nanopore reads. For the LCP06136 parental genome, the raw Nanopore reads were trimmed and assembled with Canu v1.8 [101] with the option genomeSize = 28m. The assembly obtained with Canu was polished twice with Illumina reads [31] using Pilon v1.24 [102] with the default settings. For each round of polishing, Bowtie2 [103] was used to align the Illumina trimmed reads with the assembly for polishing with a maximum length (-X) of 1000 bp. The Illumina reads were first trimmed with Trimmomatic v0.36 [104] with the following options: ILLUMINACLIP:TruSeq3-PE.fa:2:30:10 LEADING:10 TRAILING:10 SLIDINGWINDOW:5:10 MINLEN:50. Redundant contigs were removed based on a self-alignment using NUCmer v3.1 [105]. For other parental strains (LCP06037, LCP06039, LCP06043, LCP06059), available Illumina reads [31] were trimmed with Trimmomatic v0.36 [104] the same way as previously described, and assembled with SOAPdenovo2 v2.04 [106] using the corresponding default size 23 k-mer length. The assemblies obtained were then scaffolded using Ragout v2.3 with both the LCP06136 and the LCP06133 genomes as references [107]. For the 11 offspring genomes, the raw Nanopore reads were trimmed and assembled both with Canu v1.8 [101] and with Flye v2.9 [108], and then merged using Quickmerge [109]. These obtained assemblies were polished once with Illumina reads of the parental strain LCP06136 and LCP06133 [31] in the same way as for the LCP06136 parental genome. The obtained assemblies were scaffolded using Ragtag v2.1.0

[110]. We evaluated the quality of the final assemblies with Quast v5.0.2 [111] and their completeness with BUSCO v5.3.2 [112] using the eurotiales_odb10 lineage dataset. We checked the assemblies for contaminations with the GX cross-species aligner [113], part of NCBI's Foreign Contamination Screen tool suite, using the database shipped with version 0.5.0 (64,164 genome assemblies).

Transposable elements (TEs) were annotated using the REPET package (https://urgi.versailles.inra.fr/Tools/REPET). Briefly, the TEdenovo pipeline [114] was used to detect repeated elements in LCP06136 and LCP06133 (https://doi.org/10.57745/SIP7CH) genomes, and to provide consensus sequences. These consensus sequences were classified with PASTEC v1.3 [115], based on the Wicker hierarchical TE classification system [116] and manually curated. The resulting bulk library of consensus sequences was then used to annotate TE copies in the six whole genomes (namely, LCP06136, LCP06133, LCP06037, LCP06039, LCP06043 and LCP06059) using the TEannot pipeline [117]. Repeat-induced point mutations (RIP) in assemblies were detected using the web-based tool The RIPper [118].

All final assemblies were formatted (contigs being renamed and ordered in descending order of size), and repeats of the LCP06133 and LCP06136 parental genomes were masked with RepeatMasker v4.1.2 [119] after *de novo* repeat detection using RepeatModeler v2.0.2 [120]. We performed gene annotation on the masked assemblies of the parental strain LCP06136 using the Funnannotate v1.8.9 pipeline [121], with Braker v2.1.6 [122–125], which uses a combination of the *ab initio* gene predictors Augustus and GeneMark-ET [126] with NCBI blast [127] and blast+ [128]. We ran BRAKER twice, first using the BUSCO dataset of proteins eurotiales_odb10 [112] with the ProtHint pipeline [129–133], and then using the RNA-seq dataset from [134] mapped using STAR v2.5.4b [135] with the default settings. We combined the results of the two BRAKER runs with TSEBRA [136].

We studied synteny between assemblies by investigating collinearity of one-to-one whole genome alignments, using nucmer v3.1 [105], with 500 pb as the minimum match length. We visualized synteny between genomes by plotting one-to-one ortholog links using Circos v0.69-6 [137].

To determine the genetic divergence between parental strains, ANI (average nucleotide identity) values were estimated using FastANI v1.34 [138], based on the assembled genomes, for each pair of parental strains. The resulting ANI values are provided in Table 1.

We looked for potential *Starship* elements in LCP06133 (non-Roquefort) and LCP06136 (Roquefort) with the "annotate" and "sketch" modules of the Starfish pipeline [51], using the alignment profiles and protein sequences of tyrosine recombinases (referred as "captain") and auxiliary proteins included in the pipeline.

### Phenotyping

We selected four traits for their applied interest in cheesemaking that are easily measurable with precision and at high throughput, for which we assessed the phenotypes in all the isolated offspring obtained for the five analyzed crosses. The growth rate was measured using a ScanStation (Intersciences, Mourjou, France), which is a temperature-controlled incubator. We set the ScanStation at 22.5°C and it took pictures regularly, in batches of 100 Petri dishes, *i.e.,* the full capacity of the ScanStation. We spread 5 μL of standardized spore suspension (250 spores/inoculation) on each Petri dish, containing malt medium (Biokar BK045 HA) at 15 g.L$^{-1}$ in source water (Cristalline, Chelles, France), sterilized at 120°C during 15 minutes. The ScanStation took pictures of the Petri dishes right side up on a black background, for five days every 30 minutes, *i.e.,* 240 measures per strain (Fig 5D). The ScanStation estimated the colony area and diameter for each picture, by counting the number of pixels corresponding to the fungal colony. Because the colony diametral growth was near linear when plotted against days (S6 Fig), we used, as a measure of growth rate, the slope of the least square linear regression computed with the *lm* function of the base package R software v4.3.2 (http://www.r-project.org/). The coefficients of determination ($r^2$), measuring the fit of the data to a line, are presented in the S6 Table and the linear regression of an offspring with a median $r^2$ in the S6 Fig. To ensure the robustness of the linear fit, strains with coefficients of determination below 0.98 were not considered in the QTL analysis of this trait (21 out of 781 offspring phenotyped, *i.e.,* 2.6% of strains filtered out).

The colony color was measured after growth on raw sheep milk powder medium (Biocoop, Paris, France) at 21% in source water, sterilized at 105°C for five minutes, and mixed with 1.7% agar in source water, sterilized at 121°C for 15 minutes. After 13 days of growth, a picture was taken by the ScanStation under standardized conditions of light and on a white background (Fig 5F). We recorded for each strain the colony color decomposition using RGB (red, green and blue) and HSB (hue, saturation, and brightness) with ImageJ v1.52n [139] (S7 Table). In the RGB system, the levels of red, green and blue are represented by a range of integers from 0 to 255 (256 levels for each color). In the HSB system, hue ranges from 0 to 360 (0 is red, 120 is green, 240 is blue), saturation and brightness from 0 to 100% (0% saturated is neutral grey, 100% saturated is the full color; 0% brightness is black, 50% brightness is normal, 100% brightness is white). In order to obtain uncorrelated components from the RGB data, we divided each value by the sum of the three components. For each of these parameters, we recorded average values across all the colony pixels for each strain. For QTL detection, we chose *a posteriori,* among these color parameters, those whose distributions and heritabilities were optimal for QTL analysis (*i.e.,* largest and uncorrelated variance); we thus retained the RGB proportion, as well as HSB.

The lipolytic and proteolytic activities of *P. roqueforti* strains were measured *in vitro* following [31]: for each strain, 50 µl of standardized spore suspensions (ca. 2,500 spores/inoculation) were inoculated at the top of a test tube containing agar and tributyrin for lipolytic activity measure (10 mL.L$^{-1}$, ACROS Organics) or semi-skimmed cow milk for the proteolytic activity measure (40 g.L$^{-1}$, Casino). The lipolytic and proteolytic activities were estimated by the degree of compound degradation, which changes the media from opaque to translucent. We measured the distance between the initial mark and the hydrolyzed, translucent front, after 7, 14, 21 and 28 days of growth at 20°C in the dark (Fig 5E). As lipolysis curves were nearly linear when plotted against days, we estimated the lipolysis rate with a linear regression computed with the R base package function *lm*. Proteolysis profiles appeared less linear but with no clear sigmoid nor other regression patterns, so we also chose to fit linear models for more simplicity. The coefficients of determination ($r^2$), measuring the fit of the data to a line, are presented in the S6 Table and an example of a linear regression of an offspring with a median $r^2$ in the S6 Fig.

For extrolite production, we grew fungal cultures in 24-well sterile microplates containing 2 mL of yeast extract sucrose (YES) agar medium buffered at pH 4.5 with phosphate-citrate buffer and characterized by a high C/N ratio to favor extrolite production as previously described [140]. For each strain, 1 µL of a calibrated spore suspension ($10^6$ spores.mL$^{-1}$) prepared from a 7-day culture was inoculated in the center of the well. Two replicates per strain were performed for extrolite analyses. The plates were incubated at 25°C in the dark for seven days and then stored at -20°C until extrolite analysis. For extrolite extractions, we used an optimized "high-throughput" extraction method [38,45]. Briefly, 2 g-aliquots (the entire YES culture obtained from a well) were homogenized after thawing samples with a sterile flat spatula then 12.5 mL of acetonitrile (ACN) supplemented with 0.1% formic acid (v/v) was added, samples were vortexed for 30 sec followed by 15 min sonication. Extracts were again vortexed before being centrifuged for 10 min at 5000g at 4°C. The supernatants were directly collected and filtered through 0.45 µm PTFE membrane filters (GE Healthcare Life Sciences, UK) into amber vials and stored at -20°C until analyses. Extrolite detection and quantification were performed using an Agilent 6530 Accurate-Mass Quadropole Time-of-Flight (Q-TOF) LC/MS system equipped with a Binary pump 1260 and degasser, well plate autosampler set to 10°C and a thermostatted column compartment. Filtered 2 µL aliquots were injected into a ZORBAX Extend C-18 column (2.1x50 mm and 1.8 µm, 600 bar) maintained at 35°C with a flow rate set to 0.3 mL.min$^{-1}$. The mobile phase A contained milli-Q water + 0.1% formic acid (v/v) and 0.1% ammonium formate (v/v) while mobile phase B was ACN + 0.1% formic acid. Mobile phase B was maintained at 10% for 4 min, followed by a gradient from 10 to 100% for 18 min. Then, mobile phase B was maintained at 100% for 5 min before a 5-min post-time. Samples were ionized in positive (ESI+) electrospray ionization mode in the mass spectrometer with the following parameters: capillary voltage 4 kV, source temperature 325°C, nebulizer pressure 50 psig, drying gas 12 L.min$^{-1}$, ion range 100–1000 m/z. Target extrolite characteristics used for quantifications are given in S8 Table and included commercially available extrolites produced by *Penicillium* species, namely andrastin A, eremofortins A & B, (iso)-fumigaclavin A, mycophenolic acid and roquefortine C.

Andrastin A, eremofortins A & B and (iso)-fumigaclavin A standards were obtained from Bioviotica (Goettingen, Germany), while all others were from Sigma-Aldrich (St Louis, MO, USA). All stock solutions were prepared in dimethyl sulfoxide (DMSO) at $1\,mg.mL^{-1}$ in amber vials. As the PR toxin was not commercially available, a previously produced purified stock solution [34] without known concentration was used to ensure MPA [mycophenolic acid) and PR toxin were separated (as they have the same mass) as well as linearity of PR toxin quantification. For these analyses, metabolite identification was performed using both the mean retention time $\pm 1\,min$ and the corresponding ions listed in S8 Table. We used a matrix-matched calibration curve ($r^2 > 0.99$ for all extrolites except $2 > 0.96$) to confirm linearity the relation between signal area and extrolite concentration with final concentrations ranging from 1 to $10000\,ng.mL^{-1}$ according to the target metabolite and method performance was carried out as previously described [34].

## Statistical analysis on phenotypes

For phenotype distributions, we performed tests of normality with the *shapiro.test* function of the base package of R software v4.3.2 (http://www.r-project.org/). We ran tests for unimodality with the *dip.test* function of the Diptest package v0.75-7 [141] and tests for bimodality with the *bimodality_coefficient* function of the Mousetrap package v3.2.1 [142] with a threshold of 0.555 according to [143]. We performed the trimodality and amodality tests with the *is.trimodal* and *is.amodal* functions, respectively, of the LaplacesDemon v16.1.6 package [144]. We computed the average heterosis as ((F1 - MP)/ MP) * 100, F1 and MP being the trait mean values of the progeny and the parents, respectively.

We computed the matrix of Pearson and Spearman correlation coefficients with the R stats package v4.3.2 (http://www.r-project.org/). We produced the violin plots using vioplot v0.4.0 [145], ggplot2 v3.4.3 [146] and reshape2 v1.4.4 [147] R packages. We performed the principal component analysis (PCA) using the function *PCA* from the R base package, assigning trait means to missing values. Plots were drawn using FactoMineR v2.8 [148], factoextra v1.0.7 [149] and corrplot v0.92 [150] packages.

## Phenotype heritability

Narrow-sense heritability ($h^2$) for each cross x phenotype combination was computed using a linear mixed model approach implemented in the "rrBLUP" R package [151,152]. Phenotypic values were modeled in a mixed model, using "mixed.solve" as a function of a genetic relatedness matrix derived from the genotypes matrix with the function "A.mat" of the "rrBLUP" package, to estimate additive genetic variance ($\sigma_A$) and residual variance ($\sigma_e$). The narrow-sense heritability was calculated as the proportion of the total phenotypic variance explained by additive genetic effects ($h^2 = \sigma_A / (\sigma_A + \sigma_e)$). Narrow-sense heritability estimates for each cross x phenotype combination were further refined using the delete-one jackknife method to assess the stability of the estimates and compute standard errors. For each trait within each cross, individual segregants were iteratively excluded, and heritability was recalculated to derive standard errors.

Extrolite production was determined for three crosses (SxN, RxN and RxL), with two replicates for each strain; we therefore estimated broad-sense heritability ($H^2$) for these phenotypes for each cross based on variability among replicates, in addition to narrow-sense heritability as for the other phenotypes. Broad sense heritability was determined by performing a one-way ANOVA with the *lm* function of the stats v4.2.1 package in R.

## Genotyping based on indels

We genotyped 389 offspring from the cross RxN between LCP06136 (Roquefort; MAT1–1; https://www.ebi.ac.uk/ SAMEA103939766) and LCP06133 (non-Roquefort; MAT1–2; SAMEA103939763) with 200 markers with a polymorphism of amplification size between the two parental strains, due to the presence of indels (S9 Table). Primers were designed (i) to yield an amplicon size between 150–410 bp and a difference ranging from six to 20 bp between alleles, (ii) to be compatible in multiplex and (iii) to be evenly distributed along the assembly of the non-Roquefort reference genome available at the beginning of the study (FM164; EMBL accession numbers HG792015 to HG792062, [153]. Indels were detected

after a whole assembly alignment (blast v2.9.0; [127] of the two parental strains with the gap and extension penalties set to 1. After adding 250 base pairs upstream and downstream from the hit position, we performed a multi-alignment using MAFFT [154] with the whole genomes of the parental strains of the other crosses. We only kept a consensus sequence generated using consambig [96] and designed primers using Primer3 [155].

To reduce the cost of fluorescence primers, we only used four universal fluorescent primers (M13(-21), D8S1132f, D12S1090f, DYS437f), labeled with different fluorescent dyes (FAM, ATTO550, ATTO565, YAKIMA Yellow; Eurofins, Ebersberg, Germany). The specific primers attached during PCR to universal fluorescent primers thanks to a tail added to each specific primer matching an edge of the universal primers [156,157]. For each locus, the PCR amplification thus required three primers: the forward fluorescent universal primer, the forward specific primer with a 5' tail corresponding to the chosen universal primer and the regular specific reverse primer. The 200 markers were genotyped in 25 multiplex groups of 8 locus organized according to their amplicon size and their color fluorescent dye (S9 Table). For each multiplex, two separate sets of quadruplex PCR reactions were performed to minimize the interaction between primers and then pooled into octoplexes for the genotyping step. The PCR mix contained 0.16 µM of each of the fluorescent universal primer and of the reverse specified primer and 0.04 µM of the 5' tail forward primer in a final 15 µl reaction volume (2x QIAGEN Multiplex PCR Master Mix with 3 mM $Mg^{2+}$, 10x primer mix with 1.6 or 0.4 µM of each primer, 3 µl of DNA diluted 50 fold) using the same PCR cycle conditions as for the microsatellites loci previously described in the recombinant offspring detection paragraph, except that it started with a hybridization temperature of 62°C instead of 60°C. Genotyping and analysis were also done as previously described in the recombinant offspring detection paragraph.

## Genotyping by sequencing

For all five crosses, we genotyped the obtained progeny (1073 offspring in total) at the AGAP CIRAD platform with genotyping-by-sequencing using the *Ape*KI enzyme. DNAs were purified after digestion on QIAquick columns (Qiagen, Les Ulis, France) and their qualities were controlled using a TapeStation instrument (Agilent Technologies, Les Ulis, France). Sequencing was performed on an Illumina Hiseq 4000 (2x150 bp). The following numbers of recombinant offspring were genotyped by GBS: 384 for the cross RxN, 157 for the cross RxS, 185 for the cross RxL, 176 for the cross SxN and 171 for the cross LxN.

Demultiplexed reads were mapped to the LCP06136 high-quality genome for the crosses RxL and RxS and to the LCP06133 high-quality genome for the crosses RxN, LxN and SxN using *Bowtie2* v2.3.4.1 [103,158]. In *Bowtie2,* we set the maximum fragment length (-X) to 1000 and used the preset "very-sensitive-local". We used SAMtools v1.7 [159] to sort and filter out duplicate reads and reads with a mapping quality score above ten for SNP calling. Single nucleotide polymorphisms (SNPs) were called using the GATK v4.1.2.0 Haplotype Caller [160], generating a gVCF file per strain with option ploidy 2 to detect potential duplicated regions in progenies, as we had found indels with two alleles in some offspring. GVCFs were combined using GATK CombineGVCFs, genotypes with GATK GenotypeGVCFs, and biallelic SNPs were selected after filtration using GATK SelectVariants. We filtered SNPs using GATK VariantFiltration and options QUAL <30, DP<5, QD<20.0, FS>60.0, MQ<40, 0.05<AF<0.95, SOR>3.0, MQRankSum<-12.5, ReadPosRankSum<-8.0, DP>5.0 and GQ<10. We generated the genotype matrix based on the VCF file with *Bcftools* v1.11 [159,161]. Markers with no data or with no differences between the two parental genomes were filtered out. To reconstruct offspring genotypes, we performed SNP calling in the diploid mode in order to detect double copies of translocated regions. Markers corresponding to the translocated regions were in fact called as heterozygous in some offspring; they were therefore placed in offspring at the two parental locations, which were each assigned the corresponding parental allele.

After performing crosses and isolating ascospores, we detected sectors of two slightly different morphotypes in subsequent cultures of the parental strain LCP06039 aiming at phenotype measurements. However, genetic data indicated that the cross was performed with a pure strain and not a mixture: (i) the indel genotypes of the two morphotypes were assessed and were identical, and (ii) the GBS genotypes of the two morphotypes were not typed directly, but we did not

detect genome-wide segregation distortion in the progeny in GBS data for this cross, while a cross involving a mixture of strains with different genotypes should result in over-representation of the alleles by the pure parent; on the contrary, the segregation distortions detected in some genomic regions corresponded to over-representations of the strain showing two morphotypes rather than its under-representation. We therefore considered that a single strain contributed to the cross and that the two morphotypes were the result of plasticity or a few somatic mutations in a single strain. We therefore built the genetic map and ran a QTL analysis for the LxN cross. For analyses involving parental traits, and therefore potentially affected by the presence of two morphotypes (*i.e.,* for heritability estimates and transgression detection) we computed estimates using all three possible values for the LCP06039 parent strain phenotype: the means of the trait values for the two morphotypes or the value of one of the two morphotypes. This did not affect conclusions as the two morphotypes were actually very close for all trait measures.

### Experiment on the stability of genotypes across replication

As we identified a genomic region with two alleles present in multiple offspring in the RxN cross with associated QTLs, we wanted to check whether this particular genotype was stable during strain culture and across multiple replication steps. We therefore cultivated, on malt agar, the two parental strains from the RxN cross and the 11 offspring for which genomes were sequenced with Nanopore, six with alleles of both parents for this genomic region, four with the LCP06136 parental allele and one with the LCP06133 allele. We cultivated the strains for one week on malt agar and replated them on a new plate by transplanting some conidia from the edge of the colonies. We performed 19 replication steps. In the end, we re-genotyped the lines with the six indel markers present in the rearranged region.

### Linkage map and QTL detection

Obtained offspring phenotypes are presented in S10 Table and offspring genotypes are presented in S11–S15 Tables for SxN, LxN, RxN, RxS and RxL crosses respectively. The genetic map and statistical tests for QTL association were performed using the R software v3.5.1 (http://www.r-project.org/) with the ASMap v1.0-5 package [162] for the genetic map and qtl v1.60 package [163] for statistical tests. For the genetic map, we used unique segregating markers with less than 10% missing data and markers with segregation distortion with significant Chi Square p-values adjusted with Bonferroni corrections. Individuals with more than 30% of missing data were filtered out (Table 1), concerning up to 14% of the offspring of a cross. Linkage groups and marker ordering were determined with the *mstmap* function of AsMap package, with parameter "bychr" set as FALSE and a p.value of $10^{-12}$. Markers with less than 20% missing data and less than 95% of one parental allele were then reintegrated in linkage groups with the *pull.cross* function with "max.rf" and "min.lod" parameters set at 0.1 and 20, respectively. We reconstructed marker order in linkage groups with the *mstmap* function, the parameter "bychr" set as TRUE, and a p.value< $10^{-12}$. We estimated genetic distances between markers with the AsMap *quickEst* function and the Kosambi's mapping function. We compared the obtained order to the reference genome order using the *compareorder* function of the qtl package v1.6 with error probability set at 0.005. We changed the order only if it improved the LOD score and minimized the linkage group length. We represented the relationships between chromosome physical positions and genetic maps using ALLMAPS in JCVI utility libraries v1.3.6 [164], adapted to include genetic features.

We performed single QTL detection using the *scanone* function of the qtl package with the threshold at 5% being estimated with 1000 permutations. We used the *addqtl* function of the qtl package to mask the first set of QTLs and thus detect additional QTLs that may have been hidden by the main ones. We estimated confidence intervals for QTL using the *lodint* function of the qtl package with default parameters. We identified QTL interactions using the *scantwo* function with the Haley-Knott method using a threshold at 5% estimated by 1,000 permutations. We performed analyses of variance (ANOVAs) using a linear model with the *fitqtl* function of the qtl package, keeping interactions between loci only when they were significant in both *scantwo* and the full model AVOVAs. We extended QTL confidence intervals including the first

or last marker of a linkage group to the start or end of the corresponding chromosome. For comparisons of QTL interval positions between crosses, we transposed the intervals determined with the positions on the Roquefort parent genomes (for RxL and RxS crosses) onto the non-Roquefort genome. To do this, we extracted genome sequences using bedtools getfasta v2.30.0 [165] and mapped the extracted sequences onto the non-Roquefort parent genome using minimap2 v2.17-r941 [166]. The representation of the aligned QTL intervals was constructed using the karyoplotR v1.24.0 package [167].

## Supporting information

**S1 Fig. Focus on the L4 translocated region in the *Penicillium roqueforti* LCP06037 strain, the same pattern being observed for the L1 region in the LCP06039 strain.** The L4 region is subdivided into two subregions, L4.1 in red and L4.2 in blue, with different relative positions, suggesting a circular intermediate for the translocation. The cut sites for the proposed circular intermediate are indicated by purple and green arrows, respectively, and numbered 1 and 2. The fig shows the synteny of the L4 translocated region present in chromosome 1 in the LCP06037 lumber parental strain and in chromosome 4 in other parents and the proposed circular intermediate.
(TIF)

**S2 Fig. Distributions of phenotypes measured in the five progenies: (A) diametral growth in pixel per hour, (B) lipolysis rate in mm per day, (C) proteolysis rate in mm per day, (D) relative red, (E) green, (F) blue, (G) hue, (H) saturation and (I) brightness; progenies are represented along the x axis, RxL in red, RxS in beige, RxN in yellow, LxN in turquoise and SxN in pink.** The black point and line represent the mean and standard deviation, respectively, in progenies. The coloured points represent the parental values (lumber in green, non-Roquefort in blue, Roquefort in purple, silage in orange).
(TIF)

**S3 Fig. Distributions of extrolite production in three progenies: (A) PR toxin, (B) mycophenolic acid, (C) andrastin A, (D) eremofortin A, (E) eremofortin B, (F) (iso)-fumigaclavine A and (G) roquefortine C; progenies are represented along the x axis: RxL in red, RxN in yellow and SxN in pink.** The coloured points represent the parental values (lumber in green, non-Roquefort in blue, Roquefort in purple, silage in orange). All units are in $mL.L^{-1}$ except the PR toxin, which is expressed in arbitrary units.
(TIF)

**S4 Fig. Biplot of a principal component analysis with the two first components (A) and the third and fourth (B).** For every cross the progeny scores are displayed in a different color (LxN in red, RxL in brown, RxN in yellow, RxS in turquoise, SxN in pink) and with two density levels (from 0.05 to 0.10 in light color and above 0.10 in dark color). Large dots represent the barycentre of each progeny's scores using the same color code (the red one is under the yellow one). The 4-fold loadings of the phenotypes are represented by the black arrows. Small dots show the scores of the parental strains from different populations (lumber in green, non-Roquefort in blue, Roquefort in purple and silage in orange) in the labels show the cross they are involved in. The percentage in the axis labels shows the total variance explained by the component.
(TIF)

**S5 Fig. Number of phenotype classes (*i.e.,* lipolysis, proteolysis, color and extrolite production) presenting a QTL along the genome positions.** The crosses are represented one per line, with the cross ID indicated on the right, the first parent carrying the MAT1–1 mating type; the bottom line represents a pool of all crosses. The x-axis represents genomic physical positions (Mb). At the bottom, the four chromosomes of the reference genome (LCP06133) are represented by rectangles depicting genomic features: large horizontally transferred regions specific to the reference genome,

in blue, and translocated regions in the reference genome, in red. The large vertical empty red rectangles indicate pleiotropic QTL regions, with effects on multiple traits in at least one cross.
(TIF)

**S6 Fig. Example of phenotype measures and parameter estimation, for an offspring having a median $r^2$ value for the least squares regression fit: (A) growth dynamic, *i.e.,* colony diameter (in pixel numbers on pictures) as a function of time (hours), (B) lipolysis dynamics and (C) proteolysis dynamics, B and C representing the distance of the lysis front (*i.e.,* characterized by medium fading) in mm per day.** The linear regression lines are shown in blue, and their equations and coefficients of determination are given at the top.
(TIF)

**S1 Table. Result of normality (Wilk), unimodality, bimodality, trimodality and amodality test for each phenotype x cross combination.**
(CSV)

**S2 Table. Heterosis and heterobeltosis values and significance for each phenotype x cross combination.**
(CSV)

**S3 Table. Narrow sense and broad sense heritability for each tested phenotype.**
(CSV)

**S4 Table. List of identified QTLs for each cross and phenotype.**
(CSV)

**S5 Table. List of genome and GBS data used in this article.**
(CSV)

**S6 Table. Determination coefficient (Rsq) of growth, lipolysis and proteolysis linear regression for parents and offsprings.**
(CSV)

**S7 Table. Script used for color phenotype determination.**
(CSV)

**S8 Table. Method performance characteristics for metabolite quantification in YES medium.**
(CSV)

**S9 Table. INDEL list, primer sequence and amplicon size.**
(CSV)

**S10 Table. Phenotypes of parents and offsprings.**
(CSV)

**S11 Table. Genotypes of offpring from the SxN cross.** S: LCP06059 allele; N: LCP06133 allele.
(CSV)

**S12 Table. Genotypes of offspring from the LxN cross.** W: LCP06039 allele (L); N: LCP06133 allele.
(CSV)

**S13 Table. Genotype of offspring from the RxN cross.** R: LCP06136 allele; N: LCP06133 allele.
(CSV)

**S14 Table. Genotypes of offspring from the RxS cross.** R: LCP06136 allele; S: LCP06043 allele. (CSV)

**S15 Table. Genotypes of offspring from the RxL cross.** R: LCP06136 allele; W: LCP06037 allele (L). (CSV)

## Acknowledgments

TC and ECr acknowledge the E3GP3 network for facilitating collaborations and skills transfer from MF laboratory. We acknowledge the staff at the GENTYANE genotyping platform (INRAe GDEC, Clermont-Ferrand, France) for microsatellite and indel genotyping and the AGAP CIRAD platform for GBS sequencing. We gratefully acknowledge Intersciences for the loan and set-up of the ScanStation (https://www.interscience.com/en/products/real-time-incubator-and-colony-counter/). We thank Jeanne Ropars for her help in obtaining the progeny and Jean-Philippe Vernadet for help with bioinformatic analyses.

## Author contributions

**Conceptualization:** Thibault Caron, Daniel Roueyre, Michel Place, Christophe Chassard, Antoine Branca, Tatiana Giraud.

**Data curation:** Thibault Caron, Ewen Crequer.

**Formal analysis:** Thibault Caron, Ewen Crequer, Ricardo C. Rodríguez de la Vega.

**Funding acquisition:** Daniel Roueyre, Michel Place, Emmanuel Coton, Tatiana Giraud.

**Investigation:** Thibault Caron, Ewen Crequer, Mélanie Le Piver, Stéphanie Le Prieur, Sammy Brunel, Alodie Snirc, Gwennina Cueff, Adeline Simon.

**Methodology:** Thibault Caron, Ewen Crequer.

**Project administration:** Tatiana Giraud.

**Software:** Ricardo C. Rodríguez de la Vega.

**Supervision:** Daniel Roueyre, Michel Place, Christophe Chassard, Monika Coton, Emmanuel Coton, Marie Foulongne-Oriol, Antoine Branca, Tatiana Giraud.

**Writing – original draft:** Thibault Caron, Ewen Crequer, Tatiana Giraud.

**Writing – review & editing:** Thibault Caron, Ewen Crequer, Mélanie Le Piver, Stéphanie Le Prieur, Alodie Snirc, Gwennina Cueff, Daniel Roueyre, Michel Place, Christophe Chassard, Adeline Simon, Ricardo C. Rodríguez de la Vega, Monika Coton, Emmanuel Coton, Marie Foulongne-Oriol, Antoine Branca, Tatiana Giraud.

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
