## [Decision Letter · Decision Letter 0]

23 May 2024

Dear Dr Crequer,

Thank you very much for submitting your Research Article entitled 'Generation of diversity in the blue cheese mold Penicillium roqueforti and identification of pleiotropic QTL for key cheese-making phenotypes' to PLOS Genetics.

The manuscript was fully evaluated at the editorial level and by independent peer reviewers. The reviewers appreciated the attention to an important topic but identified some concerns that we ask you address in a revised manuscript. We therefore ask you to modify the manuscript according to the review recommendations. Your revisions should address the specific points made by each reviewer.

Required changes are:

-Shortening of both the introduction and the discussion sections, possibly combining the results and discussion sections; while doing this, it is essential to provide a broader context, referring to the existence or not of comparable studies in fungi and confronting the results of the present study with findings by others, if possible.

-Provide ascospore viability data

-Quantify diversity in the species and discuss it in context

-Provide accession numbers for DNA sequence data, primer sequences and progeny genotype data in supplementary files.

It is also important to go thoroughly into the question of why it was not possible to perform replicates for phenotype quantifications, as this was considered important for accuracy of the results.

We also strongly recommend consideration of the proposed changes to title and to the structure of the results section, which could improve readability and broaden the audience of the paper.

2) Upload a Striking Image with a corresponding caption to accompany your manuscript if one is available (either a new image or an existing one from within your manuscript). If this image is judged to be suitable, it may be featured on our website. Images should ideally be high resolution, eye-catching, single panel square images. For examples, please browse our archive . If your image is from someone other than yourself, please ensure that the artist has read and agreed to the terms and conditions of the Creative Commons Attribution License. Note: we cannot publish copyrighted images.

If present, accompanying reviewer attachments should be included with this email; please notify the journal office if any appear to be missing. They will also be available for download from the link below. You can use this link to log into the system when you are ready to submit a revised version, having first consulted our Submission Checklist .

PLOS has incorporated Similarity Check , powered by iThenticate, into its journal-wide submission system in order to screen submitted content for originality before publication. Each PLOS journal undertakes screening on a proportion of submitted articles. You will be contacted if needed following the screening process.

Yours sincerely,

Paula Gonçalves

Guest Editor

PLOS Genetics

Kelly Dyer

Section Editor

PLOS Genetics

Reviewer's Responses to Questions

**Comments to the Authors:**

Reviewer #1: This is a very clearly presented study on an industrially important fungus. I enjoyed reading it but have only a few comments.

1. What is the ascospore viability/germination rate for each of the five crosses? Can the viability differences among crosses be attributed to specific chromosomal structural differences such as translocations and inversions? This is an important point for downstream breeding as it can impact whether certain recombinants can be generated during meiosis.

2. I think the estimated narrow-sense heritabilities are highly problematic. The total genetic variance for a phenotype includes additive variance, dominance variance (which doesn't exist in haploids such as in the case here), and epistatic variance. The traditional "narrow-sense heritability" refers only to the contribution of additive variance to the total phenotypic variance and is always positive. I think the estimated "narrow-sense heritabilities" in the current version mostly reflect the epistatic gene interactions within each of the parental strains, not the additive effects. Otherwise, the h^2 value (at least for some traits) can't be negative.

3. Line 657: what proportion of the germinated spores were recombinants?

4. Line 824: Why can't the authors have replicates for phenotype quantifications for each progeny (and parental strains)? Replicates are crucial for accurate assessments of quantitative traits.

Reviewer #2: Understanding the genetic basis of quantitative traits is crucial for adaptation and breeding in domesticated organisms. Penicillium roqueforti, vital in blue cheese maturation, offers insights into flavor development. This species presents a low genetic diversity but the existence of different populations allows linkage mapping. By crossing cheese and non-cheese strains, the authors generated F1 progenies. In parallel, they analyzed parental genome assemblies, revealing large translocations and segregation distortions in offspring. Some regions exhibited transgressive segregation for traits relevant to cheese production. Quantitative trait loci (QTL) were identified for various traits, with pleiotropic effects observed. Several QTL corresponded to known biosynthetic gene clusters. F1 hybrids, with enhanced traits and diversity, offer promise for cheese production. This study advances understanding of rapid adaptation, uncovering convergent adaptation mechanisms.

The manuscript is pleasant to read, even if it is sometimes unnecessarily very long. The background is well presented but some parts could be shortened. The questions are clearly asked and the analyzes provide clear results, which fall within the scope of PLoS Genetics.

However, I have several points that need to be addressed:

Major points:

1. I think the title is not very appropriate. Performing linkage analysis systematically leads to an increase in genetic diversity in one way or another and is not an achievement in itself. Furthermore, this increase in diversity is not quantified but I will come back to it later. In the second part of the title, the adjective pleiotropic is perhaps intriguing. Are cheese-making traits more pleiotropic than others?

Ultimately this study lies in determining the genetic origin of cheese-making traits in Penicillium roqueforti. And this is it!

2. The authors talk about the genetic diversity and the fact this diversity is very low in this species. It would be nice to have a precise idea of this diversity. First, it would be interesting to have a precise idea of this diversity (Pi, Theta or % of divergence based on SNPs) and to compare it to other species of fungi and yeast. Second, regarding the isolates that are used in the context of linkage analyses, it would be interesting to have the pairwise genetic diversity between all the strains used. Finally, regarding the population mapping, it would also be essential to estimate this increase in diversity since the authors want to make it a punchline of the paper.

3. What is the added value of the part focusing on recombination. Have other studies been carried out on other fungi? Is it possible to make a comparison that would bring additional interest?

4. Concerning the pleiotropy of the traits studied, are they more pleiotropic than the traits studied elsewhere? If so, it would be good to make an exhaustive comparison!

Minor points:

1. the introduction is very long and could be shortened to go straight to the point.

2. the structure of the results section is somewhat surprising and confusing when reading. It probably needs to be restructured. For example, the paragraph “translocation between parental genomes” characterized the parental genomes. Why is it placed after phenotype distribution and heritability? The same goes for genetic maps, recombination and co. For reading, it would be better to have a genotype part, parental genomes, crosses, recombination then a trait part and finally the results of the linkage mapping

3. The discussion is redundant with the results section. In fact, results are simple and don’t need to be rephrase a second time.

Reviewer #3: This is an interesting manuscript showing how quantitative genetics can help to shed new lights in controlling phenotypic variation, although the QTLs intervals mapped are too large to identify the causative variants. One aspect of interest is the presence of large translocation among the lineages used in the crosses. I think this aspect is interesting and I would show it in more depth in manuscripts. Here some suggestions:

- what is the impact of translocation in gametes viability? Is gametes viability estimated? Large reciprocal translocations are expected to have a significant drop in gametes viability (e.g. 50%).

-are the surviving spore segregating the parental combinations of the translocated chromosomes or there are examples of CNV (e.g. aneuplodies)

- if CNVs are detected, are these link to phenotypic variation (e.g. can they explain the transgressive phenotypes) or are they have fitness cost?

- perhaps would also be interesting to include a more in depth comparison of the distribution of the recombination maps between the different cross combinations, e.g. CO and NCO hotspots

**Have all data underlying the figures and results presented in the manuscript been provided?**

Reviewer #1: **No: ** Accession numbers for DNA sequence data, primer sequences and progeny genotype data based on SSR and indels should be provided in supplementary files.

Reviewer #2: Yes

Reviewer #3: Yes

PLOS authors have the option to publish the peer review history of their article (what does this mean? ). If published, this will include your full peer review and any attached files.

**Do you want your identity to be public for this peer review?** For information about this choice, including consent withdrawal, please see our Privacy Policy .

Reviewer #1: **Yes: ** Jianping Xu

Reviewer #2: No

Reviewer #3: No

---

## [Decision Letter · Decision Letter 1]

16 Oct 2024

Dear Dr Crequer,

Thank you very much for submitting your Research Article entitled 'Identification of quantitative trait loci (QTLs) for key cheese making phenotypes in the blue-cheese mold Penicillium roqueforti' to PLOS Genetics.

The manuscript was fully re-evaluated at the editorial level and by independent peer reviewers. Based on the reviews, we will not be able to accept this version of the manuscript, but we would be willing to review a much-revised version. We cannot, of course, promise publication at that time.

The following aspects must be fully solved in order for the manuscript to be acceptable for publication:

The results section related with heritability needs to be revised addressing adequately all the concerns and suggestions put forward by the Reviewers' and the Section Editor's comments.

In addition, the Discussion section should be significantly revised. The changes should aim to remove redundancy with the Results section, discuss the results in a broader context, and yield a concise and easily readable text for a non-specialist audience. Detailed suggestions are given in the Section Editor's comments below.

Smaller issues that must also be addressed:

Lines 766 to 773 contain redundancies.

Accession numbers should be provided with the submission.

Should you decide to revise the manuscript for further consideration here, please provide a detailed list of your responses to the review comments and a description of the changes you have made in the manuscript. Also, please note that this will be the last revision round. Should the changes introduced in the manuscript be deemed insufficient, the manuscript will not be further considered for publication at PLOS Genetics.

If you decide to revise the manuscript for further consideration at PLOS Genetics, please aim to resubmit within the next 60 days, unless it will take extra time to address the concerns of the reviewers, in which case we would appreciate an expected resubmission date by email to plosgenetics@plos.org.

 You can use this link to log into the system when you are ready to submit a revised version, having first consulted our Submission Checklist .

PLOS has incorporated Similarity Check , powered by iThenticate, into its journal-wide submission system in order to screen submitted content for originality before publication. Each PLOS journal undertakes screening on a proportion of submitted articles. You will be contacted if needed following the screening process.

To resubmit, log into your Editorial Manager account and select the option 'Revise Submission' in the 'Submissions Needing Revision' folder.

We are sorry that we cannot be more positive about your manuscript at this stage. Please do not hesitate to contact us if you have any concerns or questions.

Yours sincerely,

Paula Gonçalves

Guest Editor

PLOS Genetics

Kelly Dyer

Section Editor

PLOS Genetics

Justin Fay

Section Editor

PLOS Genetics

**Section Editor's comments and suggestions:**

1) The heritability section needs revision. Heritability based on parent-offspring correlations is not reliable with a small number of families. Falconer and Mackay suggest >100. The manuscript has 5 crosses (families) and correlations based on 5 points is expected to be quite noisy. Broad and narrow sense heritability should be estimated using standard approaches applicable to cross: PMC4001867. The authors estimate Broad-sense heritability from repeatability of trait measurements and Narrow-sense heritability by comparing phenotypic similarity among individuals with their actual genetic relatedness. It seems like the authors have the data to do both of these.

3) The introduction is ok (but could be better), but discussion should really put the work into the context of a larger body of literature in order to be relevant/interesting to a broad audience. I think the paragraph on crossovers is quite nice/concise, but the other ones are mainly reiterating results and only put into the context of cheese literature.

Transgressive segregation : how do these results compare to others in the field (not just cheese studies), this is important for broader audience interpretation of this result. A comprehensive review is not required but there are certainly other fungal studies that have examined this issue.

Translocations and segregation biases : Are translocations commonly found in crosses of domesticated (with wild or other domesticated) organisms? Is this result expected or not and what is the implication for domestication/improvement. The context of this result is well described in relation to P. roqueforti but not other domesticated organisms.

Crossing over by chromosome and recombination rate : This section is concise and draws nice only the broader context of the result in comparison to prior studies.

QTL identification and the genomic architecture of adaptation : ok

Strain breeding in Penicillium roqueforti : quite specialized discussion should be broader or much more concise

**Reviewer's Responses to Questions**

**Comments to the Authors:**

Reviewer #1: I'm still confused the the narrow-sense heritability estimates. There are still negative values and two values greater than 100% (Supplementary Table 3). How could these be? By definition, narrow-sense heritability (or any heritability estimate) should be a value between 0% (when no genetic variation contributes to observed phenotypic variation) and 100% (when all the observed phenotypic variation could be explained by allelic differences).

Lines 770-773 are redundant.

Reviewer #2: The revised version is generally unsatisfactory. In fact, they don't answer certain questions and try to avoid solving problems.

1. Concerning genetic diversity, the author writes in the abstract that "the two populations of domesticated cheeses present very little genetic diversity", then in the author summary that "the two populations of domesticated cheeses of P. roqueforti each present very little genetic diversity…”. And then they still have an introduction of more than 3 pages and we still do not have the values of this diversity between these two populations. And what is the estimated divergence between the parents of the generated map population?

2. The introduction is still too long and it will really have to be shortened to get straight to the point. Concerning the discussion (5 pages in total), it's the same. The discussion remains essentially redundant with the results section. As it stands, it's really unpleasant to read. It's really too wordy for not much.

3. The point raised by the reviewer 1 regarding the heritability is interesting. It is in fact highly problematic and was not solved in the new version. It is also even mentioned in the responses of the authors: 'However, we agree that our narrow sense values are in some cases problematic but what causes it is more likely segregation bias which will favor one parental allele over the other, thus modifying the average phenotypic value of the offspring’. But again, they don’t propose to solve that. In addition, they wrote in the main text ' Some estimates were below zero or higher than 100% (Figure 4), but heritability estimates based on regressions between offspring and parent trait means may not be highly reliable (Steinsaltz et al., 2020)’. If not highly reliable, why using them? Not sure to understand how they deal with that - not reliable but not a big deal?

Reviewer #3: I don't have further feedback for the authors.

**Have all data underlying the figures and results presented in the manuscript been provided?**

Reviewer #1: **No: ** Most genomic and raw phenotypic data aren't presented/available. They should be publicly accessible before formal acceptance and publication.

Reviewer #2: Yes

Reviewer #3: None

PLOS authors have the option to publish the peer review history of their article (what does this mean? ). If published, this will include your full peer review and any attached files.

**Do you want your identity to be public for this peer review?** For information about this choice, including consent withdrawal, please see our Privacy Policy .

Reviewer #1: No

Reviewer #2: No

Reviewer #3: No

---

## [Decision Letter · Decision Letter 2]

28 Mar 2025

Dear Dr Caron,

We are pleased to inform you that your manuscript entitled "Identification of quantitative trait loci (QTLs) for key cheese making phenotypes in the blue-cheese mold Penicillium roqueforti" has been editorially accepted for publication in PLOS Genetics. Congratulations!

Yours sincerely,

Paula Gonçalves

Guest Editor

PLOS Genetics

Kelly Dyer

Section Editor

PLOS Genetics

Aimée Dudley

Editor-in-Chief

PLOS Genetics

Anne Goriely

Editor-in-Chief

PLOS Genetics

Comments from the reviewers (if applicable):

Reviewer's Responses to Questions

**Comments to the Authors:**

Reviewer #1: Thanks for making the revisions.

Reviewer #2: The authors have now updated the figures regarding genetic divergence. It appears that the heritability has been recalculated as was done in Bloom et al. 2023. Again, a real rewrite of the introduction and discussion going straight to the essential information would have been nice. However, the paper now appears to be acceptable.

**Have all data underlying the figures and results presented in the manuscript been provided?**

Reviewer #1: Yes

Reviewer #2: Yes

PLOS authors have the option to publish the peer review history of their article (what does this mean? ). If published, this will include your full peer review and any attached files.

**Do you want your identity to be public for this peer review?** For information about this choice, including consent withdrawal, please see our Privacy Policy .

Reviewer #1: No

Reviewer #2: No

**Data Deposition**

http://datadryad.org/submit?journalID=pgenetics&manu=PGENETICS-D-24-00330R2

**Press Queries**

---

## [Editor Report · Acceptance letter]

PGENETICS-D-24-00330R2

Identification of quantitative trait loci (QTLs) for key cheese making phenotypes in the blue-cheese mold Penicillium roqueforti

Dear Dr Caron,

We are pleased to inform you that your manuscript entitled "Identification of quantitative trait loci (QTLs) for key cheese making phenotypes in the blue-cheese mold Penicillium roqueforti" has been formally accepted for publication in PLOS Genetics! Your manuscript is now with our production department and you will be notified of the publication date in due course.

With kind regards,

Anita Estes

PLOS Genetics

On behalf of:
